# Isolation and Identification of Endophytic Bacterium B5 from *Mentha haplocalyx* Briq. and Its Biocontrol Mechanisms Against *Alternaria alternata*-Induced Tobacco Brown Spot

**DOI:** 10.3390/jof11060446

**Published:** 2025-06-12

**Authors:** Qunying Qin, Boyu Liu, Baige Ma, Xihong Wei, Yi Zhou, Zhengxiang Sun

**Affiliations:** 1College of Agriculture, Yangtze University, Jingzhou 434025, China; qqy123452025@126.com (Q.Q.); 15271728234@163.com (B.L.); m15136238290@163.com (B.M.); wxhwgt@163.com (X.W.);; 2MARA Key Laboratory of Sustainable Crop Production in the Middle Reaches of the Yangtze River (Co-construction by Ministry and Province), Hubei Key Laboratory of Waterlogging Disaster and Agricultural Use of Wetland, College of Agriculture, Yangtze University, Jingzhou 434025, China

**Keywords:** *Bacillus velezensis*, tobacco brown spot, *Mentha haplocalyx* Briq., antagonistic bacteria, antifungal mechanisms

## Abstract

The fungus *Alternaria alternata*, which causes tobacco brown spot disease, poses a serious threat to the tobacco industry. Beneficial microorganisms and their secondary metabolites have emerged as a promising green strategy for disease management. This study recovered 16 endophytic bacterial strains from *Mentha haplocalyx* Briq., a therapeutic herb. The study revealed that strain B5, with an inhibition rate of 82.76%, exhibited the highest antifungal activity against *A. alternata*. This strain exhibited broad-spectrum antifungal activity, with inhibition rates ranging from 66.34% to 87.23%. Phylogenetic analysis of 16S rDNA and *gyrA* gene sequences identified it as *Bacillus velezensis* (GenBank: PV168970 and PV173738). Further characterization revealed that strain B5 can secrete cell wall-degrading enzymes, produce IAA, and synthesize siderophores. The growth of mycelium in *A. alternata* was greatly reduced by both the ethyl acetate extract and the filtered liquid from the sterile fermentation, resulting in marked morphological abnormalities. Multiple antifungal active substances were identified through liquid LC-MS analysis. Greenhouse experiments demonstrated that the B5 fermentation broth effectively suppressed the occurrence of tobacco brown spot disease, achieving a relative control efficacy of 60.66%, comparable to that of 10% difenoconazole water dispersible granule (WDG). Additionally, strain B5 enhances plant disease resistance by activating the activities of key defense enzymes. *B. velezensis* B5 serves as a safe alternative to chemical fungicides and is highly effective at controlling tobacco brown spot disease.

## 1. Introduction

Brown spot disease in tobacco (*Nicotiana tabacum*) is believed to be associated with the fungus *Alternaria alternata*. Throughout the tobacco plant’s growth cycle, this pathogen adversely impacts the plant by inducing pathological conditions, primarily in the leaves. In addition to causing leaf infections, *A. alternata* can further exacerbate plant health issues, affecting stems, peduncles, and capsules as the infestation progresses [1]. Within the realm of agricultural pathology, the species *A. alternata* is notorious for infecting a wide range of crops. It causes diseases that degrade crop quality and reduce yield, thereby incurring substantial economic losses [2].

In China, the majority of tobacco cultivation areas currently suffer from a lack of adequate expertise in disease prevention and management strategies among local cultivators. Tobacco makers have suffered large financial losses as a result of a rising incidence of tobacco brown spot disease caused by this deficit. Chemical control is the use of chemicals, such as insecticides and fungicides, to manage plant diseases and pests. Key chemicals used in this approach include fludioxonil and cyprodinil, among others [3]. Chemical control offers advantages, such as ease of preparation and rapid effectiveness [4]. However, it also has significant drawbacks, including the development of disease resistance, the disruption of soil microbial ecosystems, and pesticide residue issues that have not yet been adequately addressed by current technologies [5,6].

Due to the numerous drawbacks associated with chemical control methods, environmentally friendly and less health-threatening biological control strategies have emerged as a focal point in research aimed at managing tobacco brown spot disease. Biological control involves the utilization of beneficial organisms and their bioactive capabilities, including their mechanisms of action and natural interactions, to suppress the development of pathogenic species [7]. In this process, biocontrol agents, particularly antagonistic microorganisms, play a pivotal role by establishing dominant microbial communities that contribute to disease suppression.

Endophytic bacteria serve as potential biocontrol agents, with numerous studies documenting their ability to enhance plant growth and fortify plant health through various mechanisms [8,9]. These endophytes are essential components of the plant’s internal microbiota, capable of inducing the expression of defense-related genes, producing secondary metabolites that inhibit or eradicate pathogens, competing with pathogens for resources and spatial niches, and disrupting pathogen virulence [10]. During long-term coevolution, the microbiome of medicinal plants regulates the production of secondary metabolites. Their endophytes can synthesize antimicrobial compounds that effectively inhibit pathogens [11,12]. For example, Sahu et al. [13] isolated eight strains of *Bacillus* spp. from *Ocimum tenuiflorum*, all of which were able to reduce the incidence of sheath rot disease to varying degrees. Iantas et al. [14] investigated the antimicrobial potential of endophytic fungi from the leaves and petioles of two Brazilian medicinal plants against phytopathogens (three fungal pathogens: *C. abscissum*, *P. citricarpa*, *F. graminearum*; and one bacterial strain: *X. citri* subsp. *citri*). From 1304 fungal isolates, they screened and identified strains exhibiting dual antifungal and antibacterial activity and further analyzed these strains. The results demonstrated that endophytic fungi from medicinal plants produce secondary metabolites with bioactivity against both plant pathogenic fungi and bacteria. The endophytes of medicinal plants possess certain antimicrobials that are important resources for screening antagonistic microorganisms against plant pathogens. In this study, we focused on the medicinal plant *Mentha* spp. (mint) as a potential source of biocontrol agents (BCAs).

*Mentha haplocalyx* Briq. has been utilized in traditional folklore medicine for over 250 years [15]. The essential oil of *M. haplocalyx* is highly valued commercially due to its high content of menthol (30–55%) and menthone (14–32%) [16]. Extracts from *M. haplocalyx* have long played a significant role in traditional medical systems for treating various ailments [17]. Notably, menthol has demonstrated a 50% growth inhibition effect on fungal pathogens, such as *Aspergillus flavus* and *A. parasiticus* [18]. In another study, menthol was shown to reduce the levels of cell wall-degrading enzymes secreted by pathogenic fungi in legumes affected by root rot pathogens, thereby contributing to a decrease in disease incidence [19]. These findings highlight the potential of *Mentha haplocalyx*. extracts as natural biocontrol agents in modern agricultural practices, offering a sustainable alternative to chemical fungicides.

The biocontrol mechanisms of biocontrol bacteria typically encompass antibiosis, competition, hyperparasitism, and the induction of resistance. Some antagonistic bacteria primarily rely on a single mechanism; whereas, many others exhibit multiple antagonistic mechanisms. For instance, *Bacillus* spp. control other organisms through multiple mechanisms, such as 1. producing cell wall-degrading enzymes (e.g., chitinase, β-1,3-glucanase, protease) to degrade fungal cell wall components (e.g., chitin, glucan, proteins); 2. synthesizing antimicrobial compounds (e.g., antibiotics, siderophores) to inhibit pathogens; 3. generating plant growth hormones (e.g., indole-3-acetic acid) to enhance plant vigor; and 4. inducing systemic plant resistance to strengthen defense against infections [20]. Proteases, β-1,3-glucanases, and chitinases are enzymes that degrade crucial components of a pathogen’s cell wall, thereby weakening it and inhibiting the growth of plant diseases [21,22]. Xu et al. [23] have found that Bacillus strains SG08-09 and SG09-12 can produce proteases and exhibit strong efficacy against tomato gray mold. Additionally, *Bacillus* spp. can inhibit the growth of cellulose-containing phytopathogenic Oomycete, such as *Phytophthora*, by secreting cellulases [24,25]. The secondary metabolites produced by endophytes include various antibiotics, such as lipopeptides (surfactins, iturins, and fengycins), polyketides (macrolactins, bacillaene, and difficidin), and aminoglycosides (butirosin), which have the potential to disrupt the structure of the cell membrane and impede the proliferation of pathogenic bacteria and fungi [26]. Understanding these multifaceted mechanisms is crucial for optimizing the application of biocontrol bacteria in sustainable agricultural practices and developing effective strategies against plant pathogens.

In this study, we targeted tobacco brown spot disease. A bacterial strain exhibiting resistance to *A. alternata* was isolated and screened from the medicinal plant *Mentha haplocalyx*. Furthermore, the strain was identified, its antifungal activity assessed, and its control spectrum determined. We also explored the strain’s antagonistic mechanism. This research offers novel microbial resources for the biological management of tobacco brown spot disease. Our research contributes to the advancement of sustainable agricultural practices by providing an environmentally benign alternative to chemical control methods for the management of tobacco brown spot disease. To our knowledge, this is the first report of an endophytic *Bacillus velezensis* strain isolated from the medicinal plant *Mentha haplocalyx* Briq., exhibiting strong antifungal activity against *A. alternata*.

## 2. Materials and Methods

### 2.1. Endophytic Bacteria Were Isolated from M. haplocalyx

*Mentha* spp. samples were collected from the West Campus of Yangtze University, Hubei, China. The samples were surface-sterilized using the following procedure: The samples were initially rinsed in sterile distilled water for 30 s. Subsequently, the specimen was immersed in 70% ethanol for 2 min, treated with 2.5% sodium hypochlorite for 2 min, and then immersed in 70% ethanol for an additional 2 min. Finally, the specimen was rinsed three times with sterile distilled water. A mortar and pestle were employed to grind the sterilized *Mentha* tissues, and the resulting suspension was serially diluted. Aliquots of the 10^−4^, 10^−5^, and 10^−6^ dilutions were incubated at 28 °C for 48 h on LB. The single colonies that arose were selected and purified by successive streaking on fresh LB agar plates. Subsequently, each purified colony was stored in glycerol depots at −80 °C [27].

### 2.2. Screening of BCAs Antagonistic to Tobacco Brown Spot Disease In Vitro

The plate confrontation method was employed to assess the antagonistic activity of the bacterial strains against fungal pathogens. The fungi to be tested were inoculated onto potato dextrose agar (PDA) plates. Mycelial discs with a diameter of 6 mm were transferred to the center of each plate after two days of incubation at 28 °C. (The fungal strains were obtained from the Fungal Specimen Room at Yangtze University, Jingzhou, Hubei, China). At four symmetrical points, each 25 mm away from the fungal plug, a paper filter disc with a circumference of 6 mm was immersed in the prepared bacterial culture solution. The controls consisted of paper filters that had been immersed in sterile distilled water. The mycelial diameter and inhibition rate were subsequently assessed after the dishes were incubated at 28 °C for two days. The entire experiment was repeated three times to ensure reproducibility, and each treatment group underwent three replicate experiments [13].Inhibition rate (%) = [(Colony diameter of control − Colony diameter of treatment)/(Colony diameter of control − Initial colony diameter)] × 100.

### 2.3. Evaluation of the Antimicrobial Spectrum of Strains

The methodology employed in this study follows the procedures described in Section 2.2. Specifically, the mycelial growth inhibition rate of the most effective antagonistic strain, B5, was assessed against six pathogenic fungi: *Phytophthora nicotianae*, *Fusarium fujikuroi*, *F. verticillioides* (formerly *F. moniliforme*), *F. graminearum*, *F. culmorum*, and *Bipolaris maydis*. All fungal strains were maintained in the Fungal Herbarium at Yangtze University, Jingzhou, Hubei Province, China. To guarantee the reproducibility and reliability of the results, each experimental treatment was implemented in triplicate.

### 2.4. Identification of the Antagonistic Strain B5

#### 2.4.1. Morphological Identification

Strain B5 was inoculated onto a Luria agar (LA) plate and incubated at 25 °C for 7 d. The morphological characteristics of the mycelium and spores were subsequently observed under a light microscope (manufactured by Ningbo Shunyu Optical Technology (Group) Co., Ltd., Ningbo, China) [28].

#### 2.4.2. Biochemical and Physiological Indicators

The physiological and biochemical characteristics of the antagonistic bacteria were analyzed using the *Manual of Systematic Identification of Common Bacteria* and *Berger’s Manual of Bacterial Identification* as reference [29]. The tests conducted included Gram staining, enzyme hydrolysis, gelatin liquefaction, sodium citrate utilization, the indole test, the V-P test, the methyl red test, and the salt tolerance test.

#### 2.4.3. Molecular Identification

To identify strain B5, we employed 16S rDNA and *gyrA* sequence amplification followed by phylogenetic analysis. The strain was inoculated into LB media and incubated overnight at 28 °C, shaking at 140 rpm. DNA was extracted using the TIANamp Bacterial DNA Kit (Tiangen Biotechnology Co., Ltd., Beijing, China). PCR amplification was carried out using universal primers that focused on the 16S rDNA [30] and *gyrA* [31]. genes. Following PCR, the amplified products were electrophoresed on a 1.0% agarose gel and sequenced at Wuhan’s Huazhong University of Science and Technology. The sequencing results were corrected, and a preliminary BLAST alignment was performed on the NCBI database (http://www.ncbi.nlm.nih.gov/BLAST, accessed on 25 October 2024). Based on the alignment results, the sequences of type isolates that are closely related were downloaded. (Table 1). Using BioEdit 7.0 software, the sequences were corrected, and *Escherichia coli* isolate L1 E925 was designated as the outgroup. The maximum likelihood (ML) method was implemented to generate a multi-gene phylogenetic tree with MEGA 5.0 software [32].

### 2.5. Effect of B5 on Tobacco Red Star Disease Control in the Greenhouse

A pot experiment was also conducted to assess the control effects of strain B5 on tobacco brown spot disease [33]. When flue-cured tobacco plants had 8 to 10 leaves, they were treated as follows: A spray method was used to evenly apply a fermentation broth containing viable strain B5 cells at a concentration of 1 × 10^8^ CFU/mL to the top of the tobacco leaves, ensuring that the liquid droplets did not drip [34]. After drying, 6 mm-diameter mycelial discs of *A. alternata* were inoculated onto each leaf, with four discs per leaf. The experiment included four treatments—CK1 (Tobacco plants are sprayed with sterile Nutrient Broth (NB) medium), CK2 (Tobacco plants sprayed with sterile NB medium and then inoculated with the pathogen), CK3 (Tobacco plants sprayed with the B5-containing fermentation broth and then inoculated with the pathogen), and CK4 (Tobacco plants sprayed with 10% difenoconazole WDG at a 2000-fold dilution and then inoculated with the pathogen). In each treatment, there were three replicates, each containing ten tobacco plants. The pot trials were carried out in a greenhouse (June–September 2024) under the following conditions: 12 h light/12 h dark cycle, 28 °C constant temperature, and RH maintained at >80% (typically >90%). The disease investigation was conducted according to the grading standard for brown spot disease once obvious disease spots appeared on CK2. The incidence, severity index, and relative control efficacy were then calculated [35].Incidence = (number of diseased leaves/total number of investigated leaves) ×100%Disease index =∑ (number of diseased leaves at all levels × disease progression)/(total number of surveyed leaves × highest disease progression) × 100Relative control effect = (disease index of the control group − disease index of the treatment group)/disease index of the control group × 100%

### 2.6. Determination of Biological Characteristics

#### 2.6.1. Extracellular Enzyme Detection

The plate method was employed to evaluate the activity of extracellular enzymes in strain B5, which includes protease, chitinase, amylase, glucanase, and cellulase. After being inoculated into LB liquid medium, the bacteria that showed positive antagonistic effects were shaken at 37 °C for the entire night at 200 rpm. Five microliters of the bacterial suspension were then poured into the middle of each matching medium. To observe the formation of a transparent zone (dissolution circle) surrounding the bacterial colonies, which is indicative of enzyme activity, the plates were subsequently incubated at 37 °C for 24 h [36].

#### 2.6.2. Siderophore Production Assessment

As previously mentioned, CAS agar plates were used to measure the bacterial strain’s generation of siderophores [37]. One microliter of bacterial fermentation broth, which had been agitated beforehand, was spotted over the center of each CAS agar plate within a laminar flow cabinet. As a control, sterile distilled water was applied to several plates. A constant–temperature incubator was then utilized to incubate the CAS plates at 28 °C. The appearance of an orange halo around the bacterial colonies indicated siderophore production. We carefully measured and recorded the diameter of the halo for further analysis.

#### 2.6.3. IAA Synthesis Capability Test

As previously mentioned, the Salkowski reagent colorimetric method was used to evaluate strain B5’s capacity to produce IAA [38]. The specific steps were as follows: Strain B5 was inoculated into LB medium that was enriched with 1 g/L tyrosine and incubated at the appropriate temperature for 3 d. The negative control was LB medium without inoculum. After 48 h of cultivation at 28 °C, 2 mL of supernatant was obtained from each culture. Each supernatant sample was then supplemented with 4 mL of Salkowski reagent and 2 droplets of orthophosphoric acid. The mixed solution was subsequently allowed to remain in the dark for 30 min to detect any color change that would suggest the presence of IAA.

#### 2.6.4. Determination of the Ability to Form Biofilm

The biofilm-forming capability of strain B5 was evaluated using glass test tubes, adhering to the protocol outlined by Sun et al. [39]. The strain was initially cultivated in LB medium at 28 °C with shaking at 130 rpm for 24 h. The bacterial suspension that resulted was subsequently attenuated to various optical densities (OD600 = 0, 0.2, 0.4, 0.6, 0.8, and 1.0) using fresh LB medium. Four microliters of the suspension was pipetted into triplicate test containers for each concentration and incubated at 28 °C for 48 h. The crystal violet staining method was employed to evaluate the formation of biofilm. In each tube, 5 mL of a 0.1% (*w*/*v*) crystal violet solution was added after incubation, and the tubes were permitted to stain for 15 min at 28 °C. Excess stain was eliminated by decanting and thoroughly washing the containers with sterile water. The presence of a purple rim on the inner wall of the tube suggested biofilm formation. To dissolve the biofilms, each tube was treated with 5 mL of 95% (*v*/*v*) ethanol, which was then gently oscillated. The absorbance of the dissolved solution was then measured at 570 nm using a UV-visible spectrophotometer (METASH, Shanghai, China), with a sample of LB medium that had no bacteria as the control.

### 2.7. Study on Non-Volatile Compounds Produced by Strain B5

#### 2.7.1. Impact of Bacterial Culture Filtrate on *A. alternata* Mycelial Growth

##### Detection of Antifungal Activity in Cell-Free Fermentation Broth

After being injected into 50 mL of LB medium, strain B5 was grown for seven days at 28 °C, while being shaken at 130 rpm on a rotary shaker. Centrifugation was used to gather the supernatant at 10,000 rpm for 15 min at 4 °C. The bacterial culture filtrate (BCF) of strain B5 was obtained by filtering the supernatant through a 0.22 µm microporous membrane. To create PDA plates with various sterile filtrate concentrations, different amounts of BCF were added to PDA medium, resulting in final concentrations of 5%, 10%, 15%, 20%, and 25% (*v*/*v*), respectively, as described in [40]. The control group received an equivalent volume of uninoculated NB medium. Three duplicates of each treatment were prepared. The center of each plate was injected with the pathogen responsible for tobacco brown spot disease, and the plates were subsequently incubated for three days at a consistent temperature of 28 °C. The inhibition rate was determined by measuring the size of the pathogen’s colony in both the treatment and control groups, following the procedures outlined in Section 2.2.

##### Observation of Hyphal Morphology by Scanning Electron Microscopy (SEM)

Following the method by Wu et al. [41], the fungal hyphae were treated with a 2.5% glutaraldehyde solution to fix them in a 1,4-dithioctane-2,3-diol-pentanoic acid (PIPES) buffer (NeoCide GB02009, Suzhou, China) overnight. After double-layer blotting paper filtration, it was rinsed twice more with the aforementioned solution and then soaked in 1% osmium tetroxide for fixation for 2 h. The fixed fungal hyphae were dehydrated in ethanol solutions of 30%, 50%, 70%, 80%, 90%, and 100% for 15 to 20 min, respectively, placed in a culture dish, and dried in a freeze dryer. The dried samples were fixed onto carbon conductive tape and subjected to ion sputter coating with gold, followed by observation under a scanning electron microscope (Tescan Vega3, Prague, Czech Republic).

#### 2.7.2. Extraction and Antifungal Activity Determination of Strain B5

Regarding the approach outlined by Mandal et al. [42], an equivalent volume of ethyl acetate was combined with 300 milliliters of sterile BCF. After being moved to a separatory funnel and given a thorough shake, the mixture was left to stand until the solution had fully separated into layers. The upper organic phase was collected after 2–3 repeated extractions, while the remaining lower aqueous phase was further extracted with ethyl acetate for a total of three extractions. The organic and aqueous phases were collected separately. A rotary evaporator was used to completely evaporate the solvent from the combined organic phases of the extractions. The dry extract that resulted was then diluted in one milliliter of HPLC-grade methanol. The ethyl acetate and aqueous fractions of the crude extract (10%, *v*/*v*) were incorporated into PDA medium. Each plate was inoculated with 6 mm diameter mycelial discs of the pathogen, which were then incubated for 7 d at 25 °C. We used PDA media that had the pathogen but not the extract as a control.

#### 2.7.3. LC-MS Analysis of Lipopeptide Active Substances of Strain B5

A high-resolution LC-MS instrument was used to further identify and evaluate the composition of the lipopeptide active compounds generated by strain B5, following their separation. The crude extract obtained after rotary evaporation was dissolved in mass spectrometry-grade methanol and subsequently diluted to 5 ppm before being dispensed into sample vials for detection. Using an LC-MS-grade acetonitrile as mobile phase B and ultrapure water as mobile phase A, a Kinetex^®^ F5 column (100 mm × 2.1 mm, 2.6 μm particle size) was used for the chromatographic separation. The LC-MS analysis of strain B5 was performed according to the protocols established by Jin et al. [43], Chen et al. [44], and Fan et al. [45]. Data interpretation and detailed analysis were conducted using SCIEX OS software version 1.7.0. The primary compositional profile of the lipopeptide active substances produced by strain B5 was determined by comparing the obtained data with existing entries in the relevant database.

### 2.8. Identification of Enzyme Activity Related to Plant Defense

In a controlled greenhouse environment, following inoculation with *A. alternata*, a total of three collections were conducted, with the third to sixth expanded leaves of each tobacco plant being sampled at 48 h intervals, counting from the top. The leaves were promptly kept at −80 °C to conduct subsequent analytical procedures. In the leaves that were subjected to various interventions, the activity levels of several defense enzymes, such as phenylalanine ammonia-lyase (PAL), peroxidase (POD), cathode (CAT), phenol oxidase (PPO), and superoxide dismutase (SOD), were assessed. Pathogen inoculation achieved the disease control, while nutrient broth (NB) inoculation achieved the healthy control. The extraction of the defense enzyme solutions and the determination of their activity were performed using a kit (Solarbio, Beijing, China) with strict adherence to the manufacturer’s instructions [46].

### 2.9. Statistical Analysis

One-way ANOVA was used to analyze the experimental data using SPSS 17.0 software (SPSS Inc., Chicago, IL, USA) [47]. The mean values of the various treatment groups were compared using Duncan’s Multiple Range Test. A significance level of *p* ≤ 0.05 was used to establish statistical significance [48].

## 3. Results and Analysis

### 3.1. Screening of Antagonistic Bacteria and Determination of Antifungal Spectrum

From the medicinal plant *Mentha* (mint), 16 types of endophytic bacteria were identified. The strain that had the most antagonistic impact was chosen and given the designation B5 through plate confrontation cultivation. In the plate confrontation assay, B5 demonstrated an inhibition rate of 82.76% against *A. alternata* (Figure 1). To evaluate the antifungal activity of strain B5 against various pathogens, six different fungal pathogens were chosen. As illustrated in Figure 2, with inhibition rates ranging from 66.34% to 87.23%, strain B5 showed antifungal activity against each of the six investigated pathogens. Notably, B5 exhibited the highest antifungal activity against the maize spot pathogen, achieving an inhibition rate of 75.23%; in contrast, its antifungal activity against the wheat scab pathogen was relatively weaker, with an inhibition rate of 59.7%.

### 3.2. Antagonistic Bacteria B5 Identification

#### 3.2.1. The Morphology of Colonies as Well as Their Physiological and Biochemical Traits

Observations of colony morphology revealed that after 1 day of streak cultivation on LB medium, strain B5 exhibited a milky white appearance on the plate, with a smooth colony surface and wrinkled edges (Figure 3a). Gram staining of strain B5 was positive (Figure 3b). Under electron microscopy, the bacterial cells appeared rod-shaped, measuring approximately 1–2 μm in length (Figure 3c). In terms of physiological and biochemical characteristics, strain B5 exhibits normal mycelial growth within the pH range of 4.5 to 8.5. Additionally, this strain can grow normally in a culture medium containing 0–10% NaCl. Strain B5 was capable of decomposing gelatin, starch, D-xylose, and D-mannitol but not L-arabinose. The Voges–Proskauer (V-P) reaction and the propionate reaction were negative for strain B5; whereas, the citrate utilization test and nitrate reduction reactions were positive (Table 2).

#### 3.2.2. Phylogenetic Analysis

The 16S rDNA and *gyrA* gene sequences were used to create a phylogenetic tree for strain B5. (Figure 4) The results indicate that strain B5 clusters with *Bacillus velezensis* B-001, with 100% bootstrap support. Considering its morphological characteristics, strain B5 is identified as *Bacillus velezensis* based on its physiological and biochemical characteristics, as well as findings from a phylogenetic study. The sequences of the 16S rDNA and *gyrA* genes of strain B5 have been deposited in the GenBank database under accession numbers PV168970 and PV173738, respectively.

### 3.3. Control Effect of B5 on Tobacco Brown Spot Disease

The results of the pot experiment (Figure 5, Table 3) demonstrate that the fermentation broth with strain B5 (1 × 10^8^ CFU/mL) exhibits a significant control effect on tobacco brown spot disease. Specifically, treatment with the fermentation broth containing viable strain B5 cells (1 × 10^8^ CFU/mL) resulted in substantial reductions in both the incidence and disease index of tobacco plants compared to the control group. The incidence of tobacco brown spot disease was reduced from 100.00% in the untreated control (CK2: NB+ *A. alternata*) to 64.81% in the B5-treated group, while the disease index was reduced from 32.92% in the control to 12.96% in the B5-treated group. This treatment achieved a relative efficacy of 60.66%, which was considerably greater than that of the untreated control group, and there was no significant difference in the relative control efficacy compared with 10% difenoconazole WDG.

### 3.4. Identification of the Biological Features of Strain B5

The assessment of strain B5 using selective media demonstrated its capacity for extracellular enzyme production, siderophore synthesis, and indole-3-acetic acid (IAA) biosynthesis. Distinct hydrolysis zones on starch, protease, glucanase, and cellulase plates confirmed the activities of amylase, protease, glucanase, and cellulase, respectively. Characteristic yellow halos around colonies on CAS medium after 7-day incubation indicated siderophore production. Additionally, pink coloration developed in the Salkowski’s reagent assay verified IAA biosynthesis. This assay involved mixing 2 mL of supernatant with 4 mL of reagent and two drops of orthophosphoric acid, following 3-day cultivation in tryptophan-supplemented PDB medium with a 30 min dark incubation (Appendix A). Using a colorimetric assay, the capacity of strain B5 to form biofilms was identified. The results demonstrated that biofilm formation was indeed possible. The amount of biofilm adhering to the tube wall increased dramatically with increasing bacterial suspension concentration, reached a peak, and then declined rapidly. The maximum biofilm formation was observed when the absorbance value of the strain B5 suspension at 600 nm was 0.6 (Figure 6).

### 3.5. Effect of Sterile Filtrate (BCF) on Mycelial Growth of A. alternata

The BCF of strain B5 significantly suppressed the development of *A. alternata*, with increasing efficacy at higher concentrations (Figure 7, Table 4). Notably, at a BCF concentration of 20%, the colony diameter of *A. alternata* was dramatically reduced to 5.88 mm, corresponding to an inhibition rate of 91%. At 25% BCF, fungal growth was completely suppressed, with no visible colonies and an inhibition rate of 100%. After treatment with BCF, changes in the structure of the fungal mycelia of *A. alternata* were looked at more closely using SEM (Figure 8). The control group’s mycelia maintained their healthy, typical structure. In contrast, mycelia treated with BCF exhibited significant morphological alterations. The SEM images revealed extensive structural deformations, including collapses, depressions, folds, and creases in the treated samples (Figure 8H). These observations suggest that the BCF inhibits fungal growth and causes severe morphological damage to the mycelia.

### 3.6. Determination of Antagonistic Activity of B5 Crude Extract

The ethyl acetate and aqueous fractions of strain B5 fermentation broth were prepared and subjected to plate assays. The findings showed that the growth of *A. alternata* was strongly suppressed by the ethyl acetate crude extract (60 mg/mL). In contrast, the aqueous fraction remaining after ethyl acetate extraction showed no inhibitory effect on *A. alternata* in the plate assay (Appendix A). Consequently, the bioactive compounds responsible for the antifungal properties are likely contained within the ethyl acetate crude extract, necessitating additional research.

### 3.7. Identification of the Constituents in Strain B5 Extracts Using LC-MS

Using SCIEX OS 1.7.0 software for LC-MS analysis, we identified various chemical components in the crude extract of strain B5. The following compounds exhibited high matching degrees and minor mass errors: Betaine (peak area = 51,550, retention time = 26.46 min, mass error = −5.0 ppm), Uridine (peak area = 436.7, retention time = 9.18 min, mass error = 0.1 ppm), Proline–Leucine Cyclopeptide (peak area = 43,080, retention time = 4.56 min, mass error = −0.5 ppm), Iturin A-7 (peak area = 3732, retention time = 7.76 min, mass error = −1.5 ppm), and Surfactin C (peak area = 415,000, retention time = 10.64 min, mass error = −0.4 ppm) (Appendix A, Table 5).

### 3.8. Impact of Strain B5 on Tobacco Defense-Related Enzyme Activity

After treatment with B5 fermentation broth, Figure 9 illustrates the variations in the activities of POD, PPO, PAL, CAT, and SOD in tobacco leaves. The activities of POD, PPO, PAL, and CAT all exhibited a rising-then-falling pattern. After using different amounts of B5 fermentation broth, the levels of these defense enzymes were much higher than those in the sick control and the healthy control. On the fourth day after root irrigation, the activities of PAL and CAT peaked; on the sixth day, however, the POD, PPO, and SOD activity peaked. In comparison to the disease control and healthy control, the PAL activity of the tobacco leaves treated with 10^8^ CFU/mL B5 fermentation broth was 101.8 U·g^−1^·min^−1^·FW, which was 35.56% and 89.50% higher, respectively. The CAT activity was 3.35 × 10^3^ U·g^−1^·min^−1^·FW, which was 135.2% and 77.24% higher than that of the disease control and healthy control groups, respectively. In comparison to the disease control and healthy control, the POD activity was 6.18 × 10^3^ U·g^−1^·min^−1^·FW, which was 114.3% and 40.13% higher, respectively. When compared to the disease control and healthy control, the PPO activity increased by 158.3% and 33.98%, respectively, to 132.47 U·g^−1^·min^−1^·FW. Finally, 138.09 U·g^−1^·min^−1^·FW was the SOD activity, which was 56.56% and 67.34% greater than the disease control and healthy control, respectively. The increase in plant defense enzyme activities with the rise in bacterial solution concentration demonstrates that the bacteria can stimulate plant defense responses, thereby facilitating the plants’ resistance to pathogen infection.

## 4. Discussion

Biological control, which utilizes microorganisms to mitigate plant diseases, is considered beneficial for plant disease management. The production and buildup of secondary compounds in the host have been greatly impacted by the microbiome of medicinal plants over long-term coevolution [49]. Therefore, bioactive substances that possess antagonistic qualities may be found in endophytes associated with therapeutic plants. For instance, Kim et al. [50] isolated the endophytic fungus *Phoma* sp. PF2 from the leaves of the Korean traditional medicinal plant *Artemisia argyi*, and after structural determination via NMR and MS, they found that phomalide A exhibited moderate antifungal activity against *Staphylococcus aureus*, while phomalide B showed weak cytotoxicity against HeLa cells, indicating that *Phoma* sp. PF2 is a potential source of bioactive secondary metabolites for drug development. This investigation identified 16 distinct endophytic bacterial strains from *Mentha haplocalyx* Briq, the source of the bacterial endophytes. Among them, strain B5, which exhibited the strongest antagonistic effect, was screened out and demonstrated antifungal activity against all six pathogenic fungi tested. The identification of strain B5 as *B. velezensis* was confirmed through morphological observation, 16S rDNA gene sequencing, and *gyrA* gene sequence analysis, aligning with the identification of *B. velezensis* BAC03. Antagonistic bacterial strains with biocontrol potential typically exhibit a wide range of antifungal activity [51,52]. Tan et al. [53] identified a strain of *B. subtilis* HZ-72 from flax rhizosphere soil [54], which inhibited the growth of seven distinct pathogenic fungi by more than 65% and demonstrated the effective prevention of flax seedling blight in potting tests. The growth of six different plant pathogenic fungi was also considerably inhibited by strain B5 in this study, with inhibition rates higher than 55%. Based on these experimental findings, strain B5 holds promise for the management of various plant diseases.

The purpose of this study was to evaluate the efficacy of live strain B5-containing fermentation broth (1 × 10^8^ CFU/mL) in controlling tobacco brown spot disease under greenhouse conditions. The results demonstrated that this fermentation broth treatment effectively reduced disease severity. These findings confirm its potential as a biocontrol agent for disease management. In a separate investigation, Xie et al. [55] reported that strain LZ88 dramatically decreased the severity of tobacco brown spot disease in greenhouse seedlings by 78.64% compared to untreated control plants. The differences in indoor preventive effectiveness observed between these studies may be attributed to the varying sources of *B. subtilis* utilized in the tests.

Strain B5 does not secrete chitinase or cellulase, but it does produce cell wall lytic enzymes, proteases, amylases, and glucosidases. Additionally, B5 can release siderophores. Similarly, *B. velezensis* F21, a biocontrol bacterium effective against watermelon wilt disease, lacks chitinase production but may secrete proteases, glucosidases, and other hydrolytic enzymes. In contrast, *B. velezensis* NKG-2, as reported by Myo et al. [56], is capable of producing siderophores, chitinase, cellulase, amylase, and β-glucosidase. Variations in strains and experimental conditions may be the cause of these disparities. Numerous related publications have linked *Bacillus*’s capacity to release extracellular hydrolases to their inhibitory effects on plant fungal infections [57,58]. Furthermore, siderophores produced by plant-beneficial bacteria are crucial for defending plants against fungal diseases [59]. These small molecular organic compounds, which have a strong affinity for Fe^3+^, can inhibit the growth of harmful fungi by maintaining iron levels below what is required for their regular metabolism [60]. Additionally, in iron-limited environments, *Bacillus* can release abundant iron chelators and siderophores to facilitate the rapid acquisition of iron ions by plants, thereby promoting healthy growth and reducing the risk of disease. In line with the findings of Feng et al. [61], who reported that four endophytic bacteria strains isolated in their study could synthesize IAA by utilizing intermediary metabolites, such as tryptamine and indole ethyl amide, strain B5 was also capable of producing IAA. Most bacteria adapt to complex ecological environments by forming biofilms. Sun et al. [39] investigated the ability of *Pseudomonas putida* A1 to colonize tomato root surfaces and utilized a colorimetric technique to assess the bacterium’s biofilm-forming capacity. Their results indicated that the concentration of the fermentation fluid increased as well, and this strain’s capacity to produce biofilms. Furthermore, *P. putida* A1 could colonize tomato roots in large numbers, with a preference for colonization near wounds. Similarly, strain B5 exhibits a high capacity for biofilm formation; as the concentration of the initial fermentation broth increases, the amount of biofilm formed by this strain first increases notably before rapidly decreasing. Preliminary speculation suggests that strain B5 may possess exceptional colonization abilities.

These secondary metabolites can stimulate plant growth, induce systemic disease resistance in plants, and directly inhibit fungal pathogens. According to Moon et al. [62], *B. velezensis* CE 100 was investigated for its ability to reduce the suppression of mycelial growth in *Phytophthora* root rot, resulting in mycelial enlargement and distortion. The sterile fermentation broth of strain B5 demonstrated potent antifungal effects against *A alternata* in this investigation. Additionally, treated samples displayed significant structural deformations, such as collapse, depression, folding, and creasing, when observed under a scanning electron microscope. The active components of strain B5’s supernatant were extracted using ethyl acetate as the organic solvent, and the bioactive compounds present in the B5 crude extract were identified through LC-MS analysis. The identified bioactive compounds, including surfactin and iturin, are well-documented for their antifungal properties and their ability to enhance plant defense mechanisms. Similar to our findings, *B. methylotrophicus* DR-08 has been reported to produce antimicrobial compounds, such as difficidin, which significantly reduces tomato bacterial wilt and inhibits the growth of a broad range of plant pathogens [63]. Specifically, fengycin is known to inhibit mycelial growth by inducing deformation, oxidative damage, and mitochondrial dysfunction in fungal cells, while surfactin enhances the activity of plant defense-related genes and enzymes, thereby contributing to improved disease resistance [64].

Plant defense-related enzymes can neutralize reactive oxygen species in plants and activate the salicylic acid (SA), jasmonic acid (JA), and ethylene (ET) signaling pathways to enhance plant resistance to pathogen infection [65,66,67]. Plants activate defense enzymes, including POD, PPO, PAL, CAT, and SOD, in response to exposure to a variety of pathogenic organisms. These enzymes are essential for the metabolism of ROS [68,69], the synthesis of secondary metabolites involved in disease resistance, including lignin, phenolic compounds, and plant defense hormones [70,71], and directly inhibit and kill pathogenic bacteria, thereby enhancing plant resistance [72]. Several studies have demonstrated a positive correlation between the disease resistance index and the activity of enzymes such as POD, PPO, PAL, CAT, and SOD, suggesting a specific defense mechanism against infections. Strain B5 substantially increased the activity of several defense-related enzymes in tobacco plants in this study, and this effect became more apparent as the concentration increased. These results are in agreement with the findings reported by Qiu et al. [66]. Similar biocontrol activity has been reported for *B. velezensis* SDTB022, a tomato rhizosphere isolate, while our study demonstrates comparable efficacy with strain B5.

The aforementioned findings imply that B5’s biocontrol activities against tobacco brown spot disease might be linked to a number of mechanisms and their combined effects. Future research is required to determine the components of B5’s antifungal chemicals and learn more about the biocontrol mechanisms behind its antimicrobial activity.

## 5. Conclusions

It is of the utmost importance to manage plant diseases in a safe, effective, and alternative manner. Biological control, which employs antagonistic microorganisms, is a sustainable and long-term method of inhibiting plant pathogens in contrast to chemical control methods. In this study, a B5 strain was isolated from *M. haplocalyx* Briq, which exhibited potent inhibitory effects against the tobacco red star disease pathogen, *A. alternata*. This strain was identified as *Bacillus velezensis* through sequencing. It exhibits substantial inhibitory effects under greenhouse conditions and possesses broad-spectrum antifungal activity against various fungal pathogens. Furthermore, it was found to enhance the expression of defense enzymes and extracellular enzymes in the host plants.

## Figures and Tables

**Figure 1 jof-11-00446-f001:**
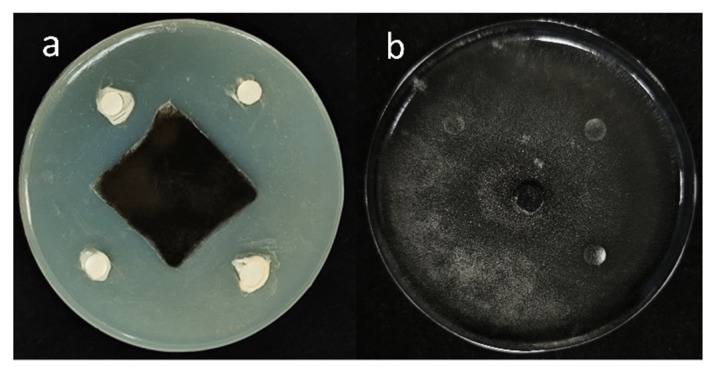
Antifungal activity of B5 against *A. alternata.* A 6 mm-diameter mycelial disc of *A. alternata* was centrally inoculated on PDA plates. Filter paper discs impregnated with B5 bacterial suspension (**a**) or sterile water (negative control, (**b**)) were aseptically placed at four equidistant positions. Significant inhibition zones surrounding B5-treated discs (**a**) demonstrated antifungal activity; whereas, confluent mycelial growth in control discs (**b**) confirmed the absence of inhibition.

**Figure 2 jof-11-00446-f002:**
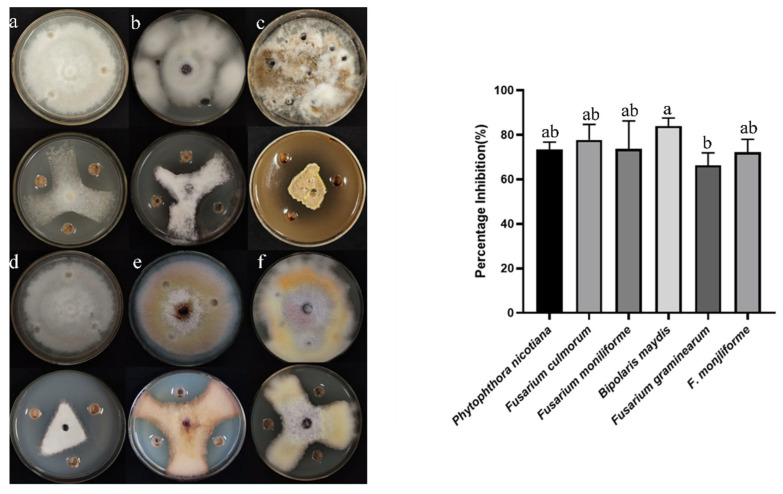
Inhibitory effects of B5 on six plant pathogenic fungi on PDA: (**a**) *P. nicotianae*, (**b**) *F. culmorum*, (**c**) *F. fujikuroi*, (**d**) *B. maydis*, (**e**) *F. graminearum*, (**f**) *F. verticillioides*. Control cultures are displayed adjacent to each test plate. Error bars indicate the standard error. Columns marked with the same letter are not significantly different (*p* ≤ 0.05, *t*-test).

**Figure 3 jof-11-00446-f003:**
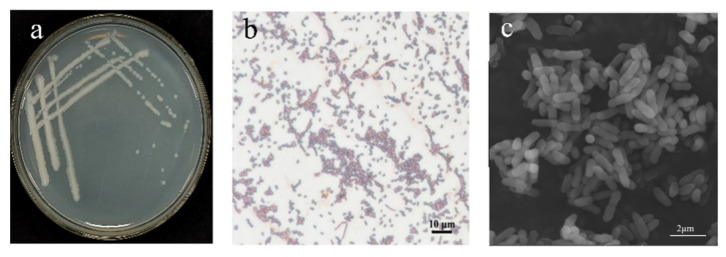
Morphological characteristics of strain B5: (**a**) colony morphology of B5; (**b**) Gram-stained cells of B5 (scale bar: 10 μm); (**c**) scanning electron microscope (SEM) images of B5 (scale bar: 2 μm/nm).

**Figure 4 jof-11-00446-f004:**
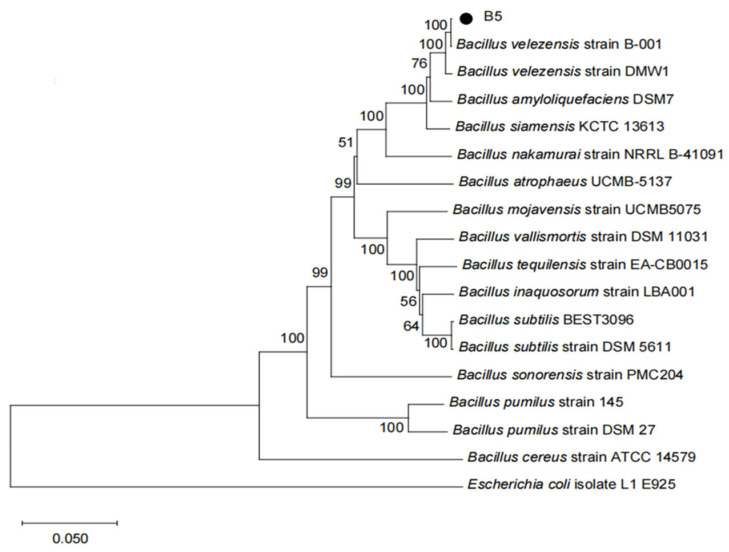
A phylogenetic tree was constructed using the neighbor-joining method based on 16S rDNA and *gyrA* gene sequences. Bootstrap values (1000 replicates) are shown at branch nodes. *Escherichia coli* isolate L1 E925 was used as the outgroup. The scale bar indicates nucleotide substitutions per site.

**Figure 5 jof-11-00446-f005:**
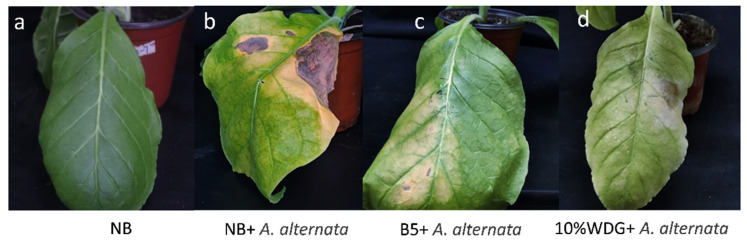
Efficacy of strain B5 against tobacco brown spot disease in a greenhouse setting. (**a**) Sterile Nutrient Broth (NB) medium control; (**b**) tobacco plants treated with sterile Nutrient Broth (NB) and then inoculated with *A. alternata*; (**c**) tobacco plants sprayed with the fermentation broth with strain B5 (1 × 10^8^ CFU/mL) followed by inoculation with *A. alternata*; (**d**) tobacco plants sprayed with 10% difenoconazole WDG at a 2000-fold dilution followed by inoculation with *A. alternata*. The images depict the comparative disease symptoms on tobacco leaves under different treatment conditions.

**Figure 6 jof-11-00446-f006:**
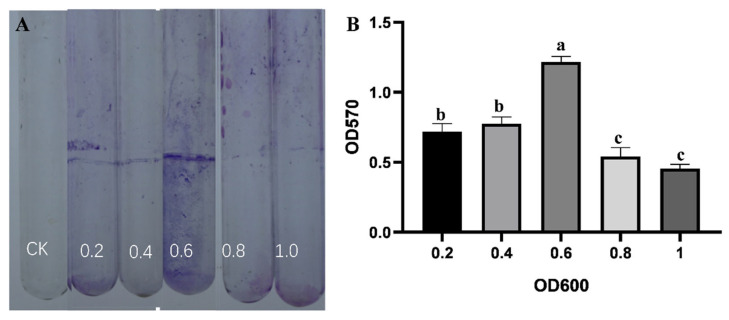
Biofilm formation analysis. (**A**) Visualization of biofilms formed on glass tubes using 0.1% crystal violet staining. CK is sterile LB broth. (**B**) Quantitative analysis of biofilm biomass under different culture densities (OD600 = 0.2, 0.4, 0.6, 0.8, 1.0) with LB medium as control. Error bars represent the standard error of the mean, and the letter denotes significant differences at *p* ≤ 0.05.

**Figure 7 jof-11-00446-f007:**
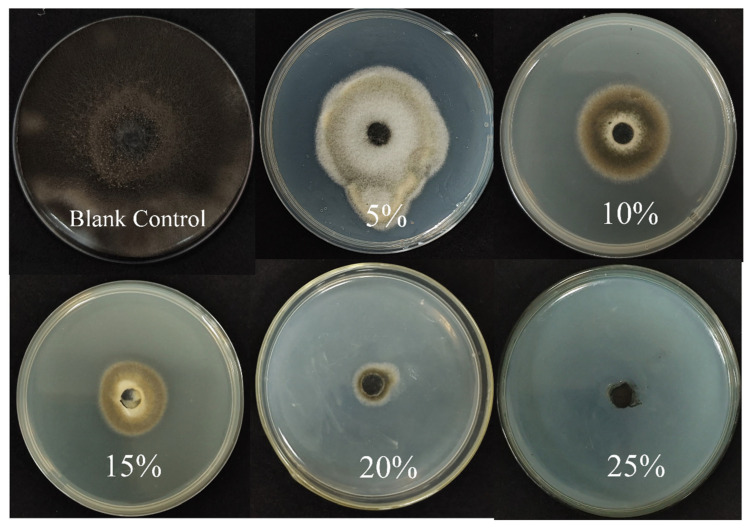
Antifungal activity of sterile filtrates against *A. alternata* at different concentrations: blank control, 5%, 10%, 15%, 20%, and 25% sterile filtrates.

**Figure 8 jof-11-00446-f008:**
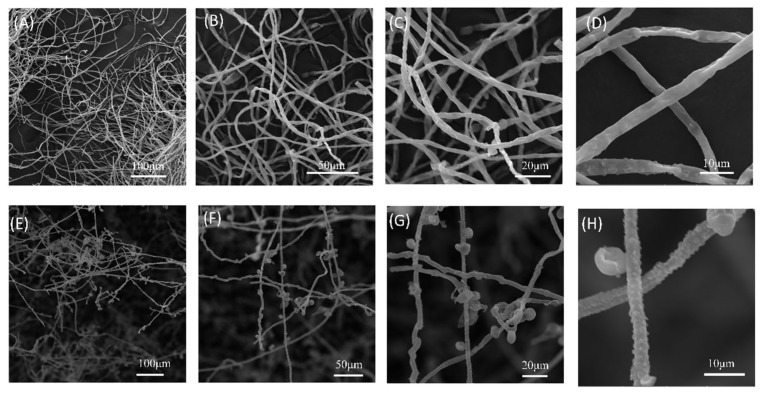
Scanning electron micrographs showing morphological alterations in *A. alternata* hyphae treated with sterile filtrate from strain B5 (1:10, *v*/*v*). (**A**–**D**) Untreated control hyphae exhibit normal morphology. (**E**–**H**) B5 filtrate-treated hyphae displaying severe structural deformations. Fungal colonies were inoculated on PDA plates, with hyphal samples collected from colony margins after 5 days of incubation at 25 °C.

**Figure 9 jof-11-00446-f009:**
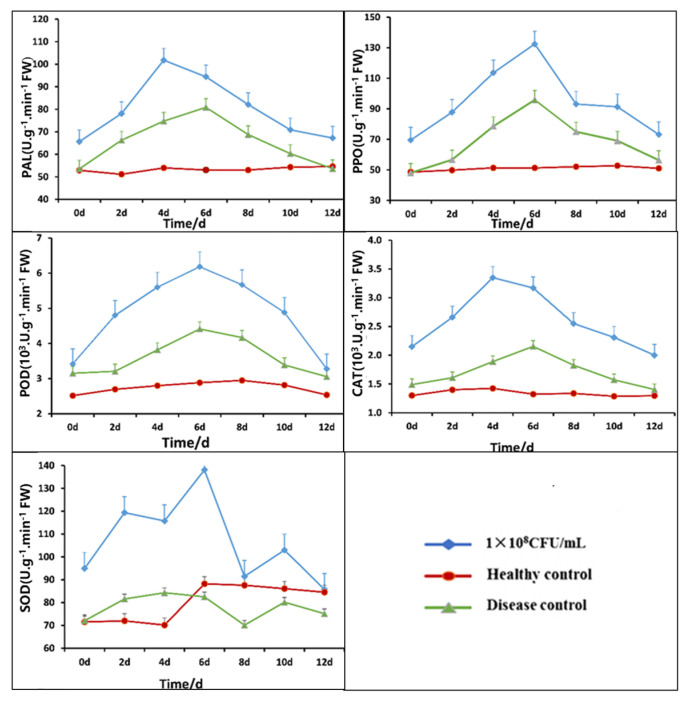
Time-course changes in tobacco defense enzymes (PAL, PPO, POD, CAT, SOD) following treatment with *Bacillus velezensis*. 1 × 10^8^ CFU/mL: 1 × 10^8^ CFU/mL B5 fermentation broth treatment; healthy control: NB treatment; diseased control: pathogenic fungi treatment.

**Table 1 jof-11-00446-t001:** Strains used in phylogenetic analysis and their corresponding GenBank accession numbers.

Entry Number			
Specific Name	Strain Name	16S	*gyrA*
*Bacillus amyloliquefaciens*	DSM7	NR_118950.1	FN597644.1
*Bacillus atrophaeus*	UCMB-5137	CP011802.1	CP011802.1
*Bacillus cereus*	ATCC 14579	NR_074540.1	CP034551.1
*Bacillus inaquosorum*	LBA001	NZ_CP127095.1	NZ_CP127095.1
*Bacillus mojavensis*	UCMB5075	CP051464.1	CP051464.1
*Bacillus Nakamura*	NRRL B-41091	LSAZ01000028.1	LSAZ01000005.1
*Bacillus pumilus*	145	CP027116.1	CP027116.1
*Bacillus pumilus*	DSM 27	CP046130.1	CP046130.1
*Bacillus siamensis*	KCTC 13613	MN176482.1	AJVF01000039.1
*Bacillus sonorensis*	PMC204	NZ_CP139190.1	NZ_CP139190.1
*Bacillus subtilis*	BEST3096	AP024622.1	AP024622.1
*Bacillus subtilis*	DSM 5611	CP120603.1	CP120603.1
*Bacillus tequilensis*	EA-CB0015	NZ_CP048852.1	NZ_CP048852.1
*Bacillus vallismortis*	DSM 11031	NZ_CP026362.1	NZ_CP026362.1
*Bacillus velezensis*	B-001	CP087957.1	CP087957.1
*Bacillus velezensis*	DMW1	NZ_CP114180.1	NZ_CP114180.1
*Escherichia coli*	L1 E925	LR883050.1	LR883050.1

**Table 2 jof-11-00446-t002:** Physiological and biochemical characteristics of antagonistic strain B5.

Strains	
Physiological and Biochemical Project	B5
V-P	−
Citrate	+
Propionate	−
D-xylose	+
L-arabinose	−
D-mannitol	+
gelatin liquefaction	+
NaCl Salt tolerance test	0–10%
PH growth	4.5–8.5
Nitrate reduction	+

Note: “+” positive result; “−” negative result.

**Table 3 jof-11-00446-t003:** Control Effect of B5 on tobacco brown spot disease.

Treatment	Incidence (%)	Disease Index	Relative Efficacy (%)
CK1			
CK2	100.00 ± 0.00 a	32.92 ± 0.58 a	
B5	64.81 ± 0.05 b	12.96 ± 1.01 b	60.66 ± 0.02 a
10% Tebuconazole WDG	68.51 ± 0.003 b	14.20 ± 0.50 b	54.86 ± 0.03 a

The values presented are the mean ± SD of three replicates; those followed by the same letter are not statistically significantly different at a *p*-value threshold of ≤ 0.05, as determined by Duncan’s Multiple Range Test.

**Table 4 jof-11-00446-t004:** The inhibition rate of different concentrations of aseptic filtrate to *A. alternata*. The pathogenic fungus was grown on PDA mixed with culture filtrates for 2 days, followed by measurement of colony diameters. Values are the mean ± SD of three replicates, and those followed by the same letter are not significantly different at P ≤ 0.05 according to Duncan’s Multiple Range Test.

Filtrate Concentration	Colony Diameters (mm)	Inhibition Rate (%)
5%	55.38 ± 1.39	23% e
10%	46.98 ± 1.21	35% d
15%	31.86 ± 1.67	56% c
20%	5.88 ± 0.30	91% b
25%	0	100% a

**Table 5 jof-11-00446-t005:** Liquid chromatography mass spectrometry (LC-MS) analysis for component identification in strain B5 crude extract.

Material Type	Molecular Formula	Ion Binding Form	Mass/Charge	Retention Time
Betaine	C_5_H_11_NO_2_	[M+H]^+^	118.0863	26.46
Uridine	C_9_H_12_N_2_O_6_	[M+H]^+^	245.0768	9.23
Cyclo(Proline–Leucine) dipeptide	C_50_H_78_N_12_O_14_	[M+H]^+^	211.1441	4.56
Cyclo(D-Phenylalanine-L-Proline	C_53_H_93_N_7_O_13_	[M+H]^+^	245.1285	5.10
Iturin A-7	C_31_H_45_O_6_P	[M+H]^+^	1071.5833	7.76
SurfactinC	C_12_H_18_N_2_O_5_	[M+H]^+^	1036.6904	10.64
Toxenutin	C_33_H_42_O_19_	[M+H]^+^	743.2393	0.89
Bacillaene	C_34_H_48_N_2_O_6_	[M+H]^+^	666.3345	4.62
Surfactin A	C_51_H_89_N_7_O_13_	[M+H]^+^	1008.6591	9.96
Surfactin B	C_52_H_89_N_7_O_13_	[M+H]^+^	1022.6748	10.31
4′,5-Trimethoxy-trans-stilbene	C_17_H_18_O_3_	[M+H]^+^	271.1329	7.52
Betaine	C_5_H_11_NO_2_	[M+H]^+^	118.0863	26.46

## Data Availability

The original contributions presented in this study are included in the article/Appendix A. Further inquiries can be directed to the corresponding author.

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
