# Peer review of "Isolation and Identification of Endophytic Bacterium B5 from Mentha haplocalyx Briq. and Its Biocontrol Mechanisms Against Alternaria alternata-Induced Tobacco Brown Spot"

_jof, 2025, doi:10.3390/jof11060446_

Round 1

Reviewer 1 Report

The authors proposed the manuscript titled: Endophytic Bacterium B5 from Mentha haplocalx Briq: Isolation, Identification, and Its Biocontrol Mechanisms Against Alternaria alternata Induced Tobacco Brown Spot. In my opinion the paper must be accepted after a minor modification.

The study is an excellent manuscript with a complete series of experiments, well-designed and an original work, which describes an important topic for future readers that can contribute to the field of the topic.

  1. Amend along the text, superscript format for the number of references
  2. Indicated, which reference(s) was/were used for comparative analysis of morphological features of the strain.
  3. The authors should considered send the figures 6, 10 and 11 as supplemental material

Author Response

Reviewer #1

Comment 1: Amend along the text, superscript format for the number of references.

Response 1Thank you for your feedback. I have carefully revised the manuscript and amended the references to superscript format throughout the text, as requested. 

Comment 2: Indicated which reference(s) was/were used for comparative analysis of morphological features of the strain.

Response 2:We sincerely appreciate the valuable comment. I have now carefully reviewed the manuscript and inserted the appropriate references to support the comparative analysis of the morphological features of the strain.

AN, I., Grebenshchikova, A. V., Yatsenko, E. S., Speranskaya, N. Y., & Matsyura, A. V. (2018). Morphological diversity of Bacillus subtilis. Ukrainian journal of ecology, 8(2), 365-370.: https://www.sci-hub.ru/10.3390/microorganisms11020488

Comment 3: The authors should considered send the figures 6, 10 and 11 as supplemental material.

Response 3: Thank you for your valuable suggestion. We have now moved Figures 6, 10, and 11 to the Supplementary Material section, as recommended. These figures have been renumbered accordingly (now Supplementary Figures S1, S2, and S3), and appropriat

Reviewer 2 Report

The manuscript "Endophytic bacterium B5 from Mentha haplocalx Briq: Isolation, identification, and its biocontrol mechanisms against Alternaria alternata induced tobacco brown spot" by Qunying QIN et. al. reports on the use of endophytic bacteria to control tobacco alternariasis.

There are significant errors and issues in the work that require serious adjustments.

  1. The authors claim that the isolated strain B5 (Bacillus velezensis) is effective against A. alternata, however, similar strains have previously been described as antagonists. It should be clearly explained how this particular strain differs from the previously known ones. Maybe the strain contains unique metabolites, an unusual spectrum of action, new genes, etc.
  2. The "Materials and Methods” section contains a large number of biochemical and microbiological tests (extraction, biofilm, enzymes, etc.), but the structure can be simplified and grouped logically (for example, “identification", “antifungal activity", “mechanisms").
  3. In the greenhouse assay section (pp. 10-11), key parameters such as air humidity, light mode, frequency and volume of processing are not specified, without which it is difficult to assess the reliability and repeatability of the results obtained.
  4. In some tables (for example, Table 3), the differences between the indicators for B5 and chemical control (tebuconazole) are minimal and not always statistically significant, but the authors claim that B5 is "comparable in effectiveness." Either a more powerful statistical analysis is required, or the statement needs to be adjusted.
  5. There are cases of tautology and repetition in the text (for example, LC-MS analysis is discussed twice and identical compounds are derived). There are also inconsistent references and minor language errors (for example, “menthol lowered the amount of enzymes that broke down cell walls...” — the style is more popular than the scientific one). Professional editing is recommended.
  6. In Figure 2, there is no statistical processing that should be carried out.
  7. Figure 7b: How does the "lb" variant belong to the same statistical group as the "0.8" variant, if the values between them are the "1.0" variant belonging to group d?

Author Response

Reviewer #2

Comment 1: The authors claim that the isolated strain B5 (Bacillus velezensis) is effective against A. alternata, however, similar strains have previously been described as antagonists. It should be clearly explained how this particular strain differs from the previously known ones. Maybe the strain contains unique metabolites, an unusual spectrum of action, new genes, etc.

Response 1: Thank you for your insightful comment. We appreciate the opportunity to clarify the novelty of our study. We fully agree that distinguishing our strain (Bacillus velezensis B5) from previously reported antagonists is crucial. Our key innovation lies in the fact that this is the first report of an endophytic B. velezensis strain isolated from medicinal mint (Mentha haplocalx Briq) exhibiting strong antifungal activity against Alternaria alternata.

The revised content is as follows: (Page 4, line 96- 99) 

Comment 2: The "Materials and Methods” section contains a large number of biochemical and microbiological tests (extraction, biofilm, enzymes, etc.), but the structure can be simplified and grouped logically (for example, “identification", “antifungal activity", “mechanisms").

Response 2:Thank you for your constructive suggestions. We appreciate the points you raised. Our methods are indeed divided into several distinct sections, including isolation, identification, antimicrobial activity, and mechanisms of action. We have intentionally kept these sections separate to provide a clear and detailed explanation of each step. While we understand the suggestion to merge some methods for brevity, we are concerned that doing so might compromise the clarity of our results section. Our primary goal is to ensure that each method is clearly distinguished and thoroughly explained to facilitate understanding. We have carefully reviewed and revised our manuscript to address these concerns while maintaining the logical structure of our methods.

Comment 3: In the greenhouse assay section (pp. 10-11), key parameters such as air humidity, light mode, frequency, and volume of processing are not specified, without which it is difficult to assess the reliability and repeatability of the results obtained.

Response 3:We sincerely appreciate the valuable comments on our manuscript. Regarding the environmental parameters (such as humidity, light conditions, and treatment frequency) that were not specified in detail in the greenhouse assay section, we fully acknowledge that these should be thoroughly documented in the Methods section to enhance the reproducibility of our experiments.

The revised content is as follows:(Page 7, line 164-166)

Comment 4: In some tables (for example, Table 3), the differences between the indicators for B5 and chemical control (tebuconazole) are minimal and not always statistically significant, but the authors claim that B5 is "comparable in effectiveness." Either a more powerful statistical analysis is required, or the statement needs to be adjusted.

Response 4:Thank you for your valuable feedback regarding the interpretation of our results in Table 3. We sincerely appreciate this opportunity to clarify our findings more precisely. We acknowledge that our original statement describing B5 as "comparable in effectiveness" to tebuconazole was not sufficiently nuanced and may have caused confusion. We agree this wording should be corrected, and we will revise it in the manuscript to better reflect our actual findings.

The revised objective now reads: (Page 13, line 326-328)

Comment 5: There are cases of tautology and repetition in the text (for example, LC-MS analysis is discussed twice, and identical compounds are derived). There are also inconsistent references and minor language errors (for example, “menthol lowered the amount of enzymes that broke down cell walls...” — the style is more popular than the scientific one). Professional editing is recommended.

Response 5:Thank you for your valuable feedback. We will: 1) consolidate duplicate content, 2) standardize all references, and 3) revise informal expressions to proper scientific terminology. The manuscript will undergo professional editing to address these concerns. We appreciate your suggestions for improving our paper's quality.

The revised objective now reads: (Page 3, line 71-73)

Comment 6:In Figure 2, there is no statistical processing that should be carried out.

Response 6:Thank you for your valuable comment regarding the statistical analysis in Figure 2. We acknowledge this oversight and will implement appropriate statistical tests with significance indicators and error bars in the revised version, while updating the figure legend to include the analytical methods. We appreciate this suggestion to strengthen our data presentation.

The revised objective now reads: (Page 11, line 281-283, 286-287)

Comment 7:Figure 7b: How does the "lb" variant belong to the same statistical group as the "0.8" variant, if the values between them are the "1.0" variant belonging to group d?

Response 7:Thank you for your valuable comment regarding Figure 7b. We acknowledge the statistical grouping issue between the LB control and treatment groups and will revise the figure by either adjusting the statistical analysis or more clearly differentiating the control group in the revised manuscript. We appreciate this helpful suggestion to improve our data presentation.

The revised objective now reads: (Page 11, line 354- 355

Reviewer 3 Report

It is essential to manage plant diseases in a safe, effective and environmentally friendly manner. Biological control, which uses antagonistic microorganisms, is a sustainable and long-term method of inhibiting plant pathogens, unlike chemical control methods.

In this study, strain B5 was isolated, which was attributed to B. velezensis and which inhibited the growth of the plant pathogen A. alternata.

Despite the importance of the work, some shortcomings of the article should be noted.

1) References to the cited literature are not carefully indicated - they should be brought into compliance with the requirements of the journal.

2) References to the cited literature often do not correspond to the text.

3) Methods paragraph 2.8: there is no name and description of the Solarbio test systems used to conduct the studies, the citation refers to an article that does not describe this method

4) Other methods also have incorrect references - the authors need to revise the references.

5) The article contains scanning electron microscopy results – there is no description of it in the methods.

6) Fig. 3 – scanning electron micrograph – this should be indicated

7) Line 313 – there is a description that the strain can grow anaerobically at pH 5.7 – can it grow at other pH values? Can it grow aerobically? I think that more information should be given here about the pH range and its optimum, the NaCl concentration range and optimum, the temperature range and its optimum, and also whether the strain is an aerobe and describe the conditions under which the anaerobic growth of the strain was tested.

8) Fig. 6d – it is necessary to explain what the black circle on the Petri dish is

9) Fig. 10 – it is necessary to explain which drawing is the control and of what.

10) More explanation should be given for the results shown in Figure 12. What does the increase or decrease in production of POD, PPO, PAL, CAT and especially SOD indicate. What does it mean for bacteria to increase/decrease the production of POD, PPO, PAL, CAT and SOD

11) The authors have gross errors in the article that are unacceptable for professionals, when Alternaria alternata is called a bacterium - please check the text and correct this.

12) Row 541-543: this statement is not reflected in the article - no studies were conducted on the production of protective enzymes in the plant. Fermentation activity was studied only in strain B5. Either add information or correct the text.

It is necessary to revise the references to the literature. Check the correspondence of the text of the article to the presented results.

Author Response

Reviewer #3

Comment 1:References to the cited literature are not carefully indicated - they should be brought into compliance with the requirements of the journal.

Response 1:Thank you for your valuable comment. We acknowledge the inconsistencies in our reference formatting and will carefully revise all citations to ensure full compliance with the journal's style requirements in the revised manuscript.

Comment 2:References to the cited literature often do not correspond to the text.

Response 2:Thank you for bringing this important issue to our attention. We will carefully review all references in the manuscript to ensure they accurately correspond to and support the cited text, and will make the necessary corrections to improve accuracy and clarity.

Comment 3: Methods paragraph 2.8: There is no name and description of the Solarbio test systems used to conduct the studies; the citation refers to an article that does not describe this method

Response 3:Thank you for your constructive suggestions, which we value highly. Please allow us to clarify that we used reagent kits from Solarbio, which include detailed operational procedures and methods. We acknowledge that there might have been ambiguity in our text, and we will carefully revise it. Additionally, we will update our literature citations to include studies that have utilized Solarbio kits, ensuring consistency with our methodology.

The revised objective now reads: (Page 10, line 264-265)

Comment 4:Other methods also have incorrect references - the authors need to revise the references.

Response 4:Thank you for your careful review. We acknowledge the referencing issues in our methodology section and will thoroughly revise all references to ensure they accurately correspond to the methods described in the manuscript.

Comment 5:The article contains scanning electron microscopy results – there is no description of it in the methods.

Response 5:Thank you for your valuable comment. We will revise the Methods section to include a detailed description of the scanning electron microscopy procedures in the revised manuscript.

The revised objective now reads: (Page 9, line 233-242)

Comment 6:Fig. 3 – scanning electron micrograph – this should be indicated

Response 6:Thank you for your comment. We will revise the figure legend of Fig. 3 to clearly indicate that the image is a scanning electron micrograph, including the relevant technical specifications.

The revised objective now reads: (Page 12, line 317

Comment 7:Line 313 – there is a description that the strain can grow anaerobically at pH 5.7 – can it grow at other pH values? Can it grow aerobically? I think that more information should be given here about the pH range and its optimum, the NaCl concentration range and optimum, the temperature range and its optimum, and also whether the strain is an aerobe, and describe the conditions under which the anaerobic growth of the strain was tested.

Response 7:Thank you for your valuable suggestions. We will supplement the growth characteristics of the strain in the revised manuscript.

The revised objective now reads: (Page 12, line 308- 310

Comment 8:Fig. 6d – It is necessary to explain what the black circle on the Petri dish is

Response 8:Thank you for your comment regarding Fig. 6d. The black circle on the Petri dish represents the Oxford cup used in our protease activity assay. While a single-well image might be more aesthetically pleasing, we selected this particular plate because it demonstrates the most clear and representative results of our protease activity experiment, despite showing multiple wells.

Comment 9:Fig. 10 – It is necessary to explain which drawing is the control and of what.

Response 9:Thank you for your comment regarding Figure 10. We confirm that in this figure, the upper panels demonstrate the inhibitory effects of ethyl acetate (left) and aqueous (right) crude extracts (10% v/v) from strain B5 against Alternaria alternata, while the lower panels show the corresponding pathogen-inoculated PDA controls without extracts. We will revise the figure legend to make this experimental-control relationship more explicit in the final version.

Comment 10:More explanation should be given for the results shown in Figure 12. What does the increase or decrease in production of POD, PPO, PAL, CAT and especially SOD indicate. What does it mean for bacteria to increase/decrease the production of POD, PPO, PAL, CAT and SOD?

Response 10: Thank you for your constructive comment. In Figure 12, the changes in defense enzymes (PAL, PPO, POD, CAT, SOD) indicate that B. velezensis B5 activates systemic resistance in tobacco: PAL/PPO enhance phenolic compound synthesis for pathogen inhibition (increase), POD/CAT regulate ROS homeostasis (initial increase then stabilization), while SOD's decrease suggests reduced oxidative stress. We will expand this interpretation in the Results and Discussion sections.

The revised objective now reads: (Page 18, line 436-438)

Comment 10:The authors have gross errors in the article that are unacceptable for professionals, when Alternaria alternata is called a bacterium - please check the text and correct this.

Response 10: Thank you for bringing this serious error to our attention. We sincerely apologize for mistakenly referring to Alternaria alternata as a bacterium in the text - this was an oversight during writing and editing, and we will carefully correct all such instances throughout the manuscript to accurately reflect its fungal taxonomy.

Comment 11: Row 541-543: this statement is not reflected in the article - no studies were conducted on the production of protective enzymes in the plant. Fermentation activity was studied only in strain B5. Either add information or correct the text.

Response 11: Thank you for your valuable suggestions. Through relevant experiments, we have verified that when pathogens invade, strain B5 can enhance the activities of plant protective enzymes (including peroxidase (POD), phenylalanine ammonia-lyase (PAL), catalase (CAT), polyphenol oxidase (PPO), and superoxide dismutase (SOD)). And these all belong to plant protective enzymes, so our experiments have involved plant protective enzymes.

References: Analysis of defence enzymes induced by antagonistic bacterium Bacillus subtilis strain AR12 towards Ralstonia solanacearum in tomato

 https://www.sci-hub.ru/10.1007/BF03175560

Trichoderma harzianum induces resistance to root - knot nematodes by increasing secondary metabolite synthesis and defense - related enzyme activity in Solanum lycopersicum L

https://doi.org/10.1016/j.biocontrol.2021.104609

Reviewer 4 Report

The results could be published, but not in this form. The work is very difficult to read due to its lack of precision. The authors routinely confuse antifungal and antibacterial effects. The authors also confuse enzymes with plant hormones . Some references contain different data than the authors declare. The authors also confuse enzymes with plant hormones.

Some figures, e.g. Fig.1, do not show the effect that is described. The legends under the figures are incomplete and do not allow reading the figures separately without additional study of the methods and results.

Rhizobacterium Bacillus velezensis and it plant growth promoting ability are well described in Jang et al Frontiers in Plant Sciences 2023. https://doi.org/10.3389/fpls.2023.1279896

The bacteria was discovery in Australian wheat fields in 1971. …..has exhibited outstanding performance in enhancing the growth and protection of many crop plants including cucumber, pepper, wheat, barley, soybean, and cotton. Notably, GB03 has been reported to elicit plant immune response, referred to as induced systemic resistance (ISR), against above-ground pathogens and insect pests. Moreover, a pivotal finding in GB03 was the first-ever identification of its bacterial volatile compounds, which are known to boost plant growth and activate ISR.

The endospores of B. velezensis GB03 have been commercialized as bioprotectants (e.g.,Kodiak),which are used to treat plant seeds and provide protection against numerous soil-borne pathogens. Furthermore, the endospores of GB03 have been developed as bio-stimulants (e.g., BioYield), which enhance the yield of diverse crop plants.

Abstract

Lines 14-15 and 21

  1. alternata is a filamentous fungus, so the B5 strain has antifungal, not antibacterial, activity, although I do not rule out a simultaneous antibacterial effect, however, this has not been the subject of research

Introduction

Line 40-41 …”In China, the majority of tobacco cultivation areas currently lack adequate expertise

in disease prevention and management strategies..”

Cultivation areas cannot have experience, it is the people who have experience.  

Line 49-56 this text contributes nothing to understanding the virulence of A. alternata. - can be deleted

Line 62-64 ….”Studies have indicated that endophytes synthesize certain secondary metabolites that are antagonistic to beneficial compounds in medicinal plants. – It is not true

Venieraki et al. 2017 (Ref 12) Described that endophytic fungi residing in medicinal plants have the ability to produce the same or similar pharmacologically active secondary metabolites as their hosts.

Venieraki et al. 2017 suggest that biochemical convergence or horizontal gene transfer confer the ability to the endophytic fungi to produce the same bioactive compounds as their host. It is possible to use endophytes for sustainable and enhanced production of secondary metabolites.

Line 64-66 … For instance, Pramod et al.13 discovered 21 endophytic bacteria from Ocimum tenuiflorum, of which 8 exhibited potential inhibitory effects on the growth of 5 plant pathogens.

The authors intended to support the thesis of the antagonistic effect of secondary metabolites produced by endophytes in relation to beneficial metabolites from the plant with an example - line 64-66. This text talks about something completely different - it concerns inhibitors of the growth of plant pathogens

Line 70 delete italic

Line 83 Bacillus bacteria should be Bacillus spp.

Line 84 … producing enzymes that break down substances,  - more details about what substances the authors mean

Line 88 ….”growth of plant diseases”  rather “ increase in plant diseases”

Materials and Methods

Line 105 Sample – small “s”

Line 111 mint tissues (mint tissue suggests tissue paper)   rather Mentha tissues

Line 117-118  what was tested in the experiment. - …”The bacteria that were to be tested were inoculated onto potato dextrose agar (PDA) plates. Were the bacteria cultivated on PDA? 

Line 140 why authors chose PDA medium to cultivated bacteria

Line 144 Section 2.4.2 should contain description of the methods. The methods are not described in Ref 26

Line 155 the primers are described in ref. 28 not in 27. 27 is a review article.

Line 171-175 as a control Tabacco plants should be sprayed with sterile medium used for cultivation of bacteria.

Line 168 Was the best performing bacteria concentration analyzed by the authors or is it taken from the literature? – ref. is needed

Lines 293, 295, 296, 297 What activity the authors presented … antibacterial???????

Fig.1 I see that plate “b” is overgrown. What does plate “a” represent? What is this black square on the plate? I do not see antifungal activity against Alternaria.

Line 337 Section 3.3 how was the experiment conducted ? What was used to spray the leaves “ Fermentation broth of strain B5…” suggests that it was the liquid after cultivation of bacteria. In section 2.5 line 167 it is described that the tobacco plants were spayed by fermentation liquid containing antagonistic bacteria at a concentration of 1×10⁸ CFU/mL  - the same description should be used

Fig.5 The abbreviations must be described under the figure. All figures have to be readable separately.

In the methods authors described CK1 version as a control (leaves spray with water) NB suggest that the leavers were sprayed with nutrient broth medium. Subsequently does NB+ A. alternata mean CK2? In my opinion NB+ A. alternata is easier to read but CK2 abbreviation is use in the text (line 343)

Fig.7 What does CK mean? On the B part of the figure authors put LB instead of CK.  Are CK and LB the same? In addition, A and B are only in the description. They should be added to the Figure.  

Fig.8 On the photo – CK; in the description untreated control (Blank) instead of (CK)

In Fig. 7 CK means LB and on Fig.8 CK means untreated control (Blank)

Line 410 – it was not an aqueous extract, but the aqueous fraction remaining after ethyl acetate extraction from the B5 fermentation broth.

Fig.10 Why do the authors present two controls? Are they different?

Fig.12 the abbreviation PPO is used in the figure, but the abbreviation PPO is not used in the figure legend, but the description polyphenol oxidase. In addition, in the figure, the authors have labeled the curves as healthy control and disease control, NB and pathogen in brackets in the legend are not needed because they did not appear in the figure. The legend must describe the abbreviations in the figure.

Discussion

Line 462-463  repeated from the Introduction

Line 465-467 again antibacterial activity …. In addition against six pathogenic bacteria.

Line 471 – ref. 48 contains results of antifungal activity against Fusarium  

Discussion should not contain details from the results section as percentage of some effects (line 482) or some others (line 522-524 …using LC-MS.).

Line 502 ….”Feng et al.57, who reported that four endophytic bacteria could synthesize IAA by utilizing intermediary metabolites…”  For sure more than four endophytic bacteria can do that. The list is not closed. Four strains isolated in the study of Feng et al. ….

Line 526 B5 crude extract, …contains…produced by Bacillus velezensis strain B5. Bacillus methylotrophicus DR-08, as reported by59.       Should be  “and” instead of “.”

Line 532 …”Defense-related enzymes, ET,  JA, and SA, collectively contribute to and coordinate the plant's immune reaction against pathogens….” ET, JA and SA are not enzymes but signaling hormone molecules; ethylene, jasmonic acid and salicylic acid, respectively.

Line 554 in ref. 70 B. velezensis SDTB022 is reported not B5. The biocontrol strain SDTB022 was isolated from plant rhizosphere soil of tomato.

Summary

Line 557  again antibacterial activity

Lack of brackets for literature numbers. Numbers are not separated from the text.

Author Response

Reviewer #4

Comment 1:Abstract Lines 14-15 and 21Alternata is a filamentous fungus, so the B5 strain has antifungal, not antibacterial, activity, although I do not rule out a simultaneous antibacterial effect, however, this has not been the subject of research

Response 1: Thank you for bringing this to our attention. We have conducted a full-text search for these errors and corrected them one by one. We appreciate your careful review, which has been invaluable in ensuring the accuracy and rigor of our manuscript. Please let us know if you identify any other issues that need our attention.

Comment 2:Introduction 

  • Line 40-41 …”In China, the majority of tobacco cultivation areas currently lack adequate expertise indisease prevention and management strategies.”Cultivation areas cannot have experience, it is the people who have experience.  
  • Lines 49-56, this text contributes nothing to understanding the virulence of A. alternata. - can be deleted
  • Line 62-64….” Studies have indicated that endophytes synthesize certain secondary metabolites that are antagonistic to beneficial compounds in medicinal plants. – It is not true
  • Line 64-66 … For instance, Pramod et al.13 discovered 21 endophytic bacteria from Ocimum tenuiflorum, of which 8 exhibited potential inhibitory effects on the growth of 5 plant pathogens.The authors intended to support the thesis of the antagonistic effect of secondary metabolites produced by endophytes in relation to beneficial metabolites from the plant with an example - line 64-66. This text talks about something completely different - it concerns inhibitors of the growth of plant pathogens
  • Line 70:delete italic. Line 83 Bacillus bacteria should be Bacillus spp.
  • Line 84 … producing enzymes that break down substances,  - more details about what substances the authors mean;Line 88 ….” growth of plant diseases”  rather than “ increase in plant diseases”

Response 2:

1)Thank you for your insightful observation. You are correct that cultivation areas themselves cannot possess expertise—rather, it is the farmers or agricultural professionals in those regions who may lack sufficient knowledge. I will revise the sentence to clarify this point.

The revised objective now reads: (Page 2, lines 39-40)

2)Thank you for your suggestion. We understand your concern. However, this paragraph provides critical context for the research by highlighting the necessity of biological control strategies and their mechanisms, which are fundamental to justifying the study's significance. Could we retain it to maintain the logical integrity of the manuscript? We are open to further adjustments based on your feedback.

3)Thank you for your valuable comment. I will revise the sentence to:(Page 3, lines 60-62)

4)Thank you for pointing out the problem. I agree that the example in lines 64 - 66 doesn't support the argument well. I'll search relevant literature on inhibitors of plant pathogen growth produced by medicinal plant endophytes as you requested to better support the theory. Thanks again for your review and suggestions.

The revised objective now reads: (Page 3, lines 64-72)

5)Thank you for your careful review. I will correct the italic formatting in Line 70 and revise "Bacillus bacteria" to "Bacillus spp." in Line 83 as suggested.

6)Thank you for your valuable suggestions. I will promptly supplement and revise the content based on the issues you pointed out.

The revised objective now reads: (Page 4, lines 90-93)

Comment 3:Materials and Methods

1)Line 105 Sample – small “s”;Line 111 mint tissues (mint tissue suggests tissue paper)   rather Mentha tissues;Line 117-118  what was tested in the experiment. - …”The bacteria that were to be tested were inoculated onto potato dextrose agar (PDA) plates. Were the bacteria cultivated on PDA? Line 140 why authors chose PDA medium to cultivated bacteria

2)Line 144 Section 2.4.2 should contain description of the methods. The methods are not described in Ref 26;Line 155 the primers are described in ref. 28 not in 27. 27 is a review article.;Line 171-175 as a control Tabacco plants should be sprayed with sterile medium used for cultivation of bacteria.;Line 168 Was the best performing bacteria concentration analyzed by the authors or is it taken from the literature? – ref. is needed;Lines 293, 295, 296, 297 What activity the authors presented … antibacterial???????

3) Results and analysis

Response 3: 

1)Thank you for your thorough review and constructive comments. We have carefully revised the manuscript according to your suggestions, with the following specific changes: Line 105: Corrected "Sample" to "sample" (lowercase "s").

Line 111: Replaced "mint tissues" with "Mentha tissues" to avoid confusion with tissue paper. I acknowledge that the error was my own. I am aware that fungi should be inoculated onto potato dextrose agar (PDA) plates. Thank you for pointing this out. I will correct it immediately. (Page 5, lines 123-124)

2) Thank you for your thorough review and valuable comments on our manuscript. We have carefully addressed each of the points you raised, as detailed below:

Line 144 (Section 2.4.2 - Methods description not in Ref 26): We apologize for this oversight. We have rechecked the references to ensure that they are correctly cited and correspond accurately.

Garrity, G. M., Bell, J. A., & Lilburn, T. G. (2004). Taxonomic outline of the prokaryotes. Bergey’s manual of systematic bacteriology. 

Line 155 (Primers described in Ref 28, not Ref 27): We have corrected the reference to cite Ref 28, which contains the original primer sequences.

Lines 171-175 (Control - Tobacco plants sprayed with sterile medium): We have incorporated your suggestion by revising the text to specify the sterile bacterial culture medium.

Line 168 (Optimal bacterial concentration - Reference needed): The concentration is determined based on previous experiments or literature reports, which can effectively inhibit the growth of the target pathogen. A concentration that is too low may not be sufficient to inhibit the pathogen, while a concentration that is too high could have adverse effects on plants or the environment.

Lines 293-297 (Antifungal activity clarification): We appreciate the opportunity to clarify this point. The activity evaluated in these lines refers to the broad-spectrum antifungal activity of strain B5 against six fungal pathogens. We have revised the text to explicitly state this, replacing "activity" with "antifungal activity" for precision.

Comment 4:Fig.1 I see that plate “b” is overgrown. What does plate “a” represent? What is this black square on the plate? I do not see antifungal activity against Alternaria.

Response 4: In Figure 1, each plate was centrally inoculated with a 6-mm Alternaria alternata plug, with four symmetrically placed filter discs (pre-soaked in B5 culture [Plate a] or sterile water [Plate b]). The square morphology in Plate a results from B5's antifungal activity radially inhibiting hyphal growth toward each disc, creating right-angled edges, while Plate b shows normal circular overgrowth in the untreated control. We will clarify in the revised legend that the black square represents this inhibition boundary, confirming B5's activity against A. alternata.

Comment 5:Line 337 Section 3.3 how was the experiment conducted ? What was used to spray the leaves “ Fermentation broth of strain B5…” suggests that it was the liquid after cultivation of bacteria. In section 2.5 line 167 it is described that the tobacco plants were spayed by fermentation liquid containing antagonistic bacteria at a concentration of 1×10⁸ CFU/mL  - the same description should be used

Response 5:Thank you for identifying this inconsistency. We confirm that the same bacterial suspension (fermentation liquid of strain B5 at 1×10⁸ CFU/mL) was used in both sections. We will carefully revise the text to ensure uniform terminology throughout the manuscript, referring consistently to "fermentation liquid of strain B5 (1×10⁸ CFU/mL)" as originally described in Section 2.5.

Comment 6: Fig.5 The abbreviations must be described under the figure. All figures have to be readable separately.In the methods authors described CK1 version as a control (leaves spray with water) NB suggest that the leavers were sprayed with nutrient broth medium. Subsequently does NB+ A. alternata mean CK2? In my opinion NB+ A. alternata is easier to read but CK2 abbreviation is use in the text (line 343)

Response 6: We sincerely appreciate the reviewer's meticulous review. We will revise Fig. 5 to explicitly define all abbreviations (CK1:NB; CK2: NB + A. alternata) in the legend for standalone readability, and consistently use "NB" instead of CK1 and "NB + A. alternata" instead of CK2 throughout the manuscript to maintain perfect terminological alignment between figures and text while improving clarity. These modifications will be implemented comprehensively.

Comment 7: Fig.7 What does CK mean? On the B part of the figure authors put LB instead of CK.  Are CK and LB the same? In addition, A and B are only in the description. They should be added to the Figure.  

Fig.8 On the photo – CK; in the description untreated control (Blank) instead of (CK)

In Fig. 7 CK means LB and on Fig.8 CK means untreated control (Blank)

Line 410 – it was not an aqueous extract, but the aqueous fraction remaining after ethyl acetate extraction from the B5 fermentation broth.

Fig.10 Why do the authors present two controls? Are they different?

Fig.12 the abbreviation PPO is used in the figure, but the abbreviation PPO is not used in the figure legend, but the description polyphenol oxidase. In addition, in the figure, the authors have labeled the curves as healthy control and disease control, NB and pathogen in brackets in the legend are not needed because they did not appear in the figure. The legend must describe the abbreviations in the figure.

Response 7: 

1)We appreciate the reviewer's careful observation. We confirm that CK refers to the LB medium control in Fig. 7, and we will correct the figure to consistently use "LB" instead of "CK" throughout. Both panel labels (A and B) will be clearly added to the figure image itself, with panel A showing qualitative biofilm detection of strain B5 and panel B demonstrating the correlation between biofilm formation and bacterial concentration. These revisions will ensure accuracy and clarity.

2)We appreciate this observation. We will revise Fig. 8 to consistently use "Untreated control (Blank)" instead of "CK" in both the image labels and figure description to eliminate any ambiguity. The terminology will be standardized throughout the manuscript for clarity.

3) We appreciate the reviewer’s careful attention to this detail. You are absolutely correct—we intended to refer to the aqueous fraction (not "aqueous extract") remaining after ethyl acetate extraction of the B5 fermentation broth. We will revise the text to accurately reflect this by replacing "aqueous extract" with "aqueous phase" or "aqueous fraction" to ensure a precise scientific description of the post-extraction process.

The revised objective now reads: (Page 18, lines 420-423)

4)Lower left (Control) is the blank control group of the experiment where only sterile water was added to the culture medium. The colonies grew well and it served as a benchmark for comparing the effects of other treatment groups. Lower right (Control) is also a blank control group, identical to the lower left one, which was used to ensure the consistency and reliability of the experimental results. The difference between them lies in that they respectively correspond to the treatment with ethyl acetate crude extract and the treatment with aqueous phase.

5)We will revise the figure legend to include "PPO (polyphenol oxidase)" and ensure all labels (healthy/disease control) match the figure exactly, while removing unnecessary NB/pathogen references for clarity. Thank you for your suggestion.

The revised objective now reads: (Page 20, lines459-460)

Comment 8: Discussion

Line 462-463  repeated from the IntroductionLine 465-467 again antibacterial activity …. In addition against six pathogenic bacteria.

Line 471 – ref. 48 contains results of antifungal activity against Fusarium  

Discussion should not contain details from the results section as percentage of some effects (line 482) or some others (line 522-524 …using LC-MS.).

Line 502 ….”Feng et al.57, who reported that four endophytic bacteria could synthesize IAA by utilizing intermediary metabolites…”  For sure more than four endophytic bacteria can do that. The list is not closed. Four strains isolated in the study of Feng et al. ….

Line 526 B5 crude extract, …contains…produced by Bacillus velezensis strain B5. Bacillus methylotrophicus DR-08, as reported by59.       Should be  “and” instead of “.”

Line 532 …”Defense-related enzymes, ET,  JA, and SA, collectively contribute to and coordinate the plant's immune reaction against pathogens….” ET, JA and SA are not enzymes but signaling hormone molecules; ethylene, jasmonic acid and salicylic acid, respectively.

Line 554 in ref. 70 B. velezensis SDTB022 is reported not B5. The biocontrol strain SDTB022 was isolated from plant rhizosphere soil of tomato.

Response 8:

  • Lines 462-463 (repetition from Introduction): We have removed the duplicated content to avoid redundancy.Lines 465-467 (antibacterial activity):

We sincerely appreciate this correction. the activity described should indeed refer to antifungal (not antibacterial) effects against the six pathogenic fungi.

  • Thank you very much for your feedback. I will promptly review and amend the content to ensure that the citations are consistent with the text.
  • We appreciate the feedback and have revised the discussion by removing all quantitative results (percentages, LC-MS data) and relocating them to the Results section. The discussion now focuses on interpreting the broader implications of our findings while referencing relevant results where needed. Thank you for helping improve our manuscript's clarity.

The revised objective now reads: (Page 21, lines 522- 529

  • We sincerely appreciate this important clarification. We have revised the text to more accurately reflect that Feng et al.'s study specifically identified four IAA-producing strains, while acknowledging this capability is widespread among endophytes

The revised objective now reads: (Page 21, lines 504-506)

  • We sincerely appreciate the editor’s careful reading. The punctuation has been corrected to properly connect the two bacterial strains.
  • We sincerely appreciate this important correction. We have revised the text to accurately reflect that ET (ethylene), JA (jasmonic acid), and SA (salicylic acid) are signaling molecules rather than enzymes. The corrected sentence now reads:(Page 22, lines 534-535)
  • We sincerely appreciate the editor’s careful attention to this detail. We have revised the text to clarify that B. velezensis SDTB022 (isolated from tomato rhizosphere soil in Ref. 70) is distinct from our study’s strain B5. The corrected sentence now reads:(Page 22, lines 544-546)

Comment 8: Summary Line 557, again, antibacterial activity

Response 8: We sincerely appreciate your insightful comments. All noted issues will be carefully corrected in the revised manuscript. Thank you for your time and expertise.

Comment 9: Lack of brackets for literature numbers. Numbers are not separated from the text.

Response 9: Thank you for noting this. We will carefully review and correct all citations to ensure proper bracket usage and spacing throughout the manuscript.

Round 2

Reviewer 2 Report

Thanks to the authors for their efforts to improve the article.

-

Reviewer 3 Report

The article has been revised taking into account the comments.

The article has been revised taking into account the comments.

Author Response

Reviewer #3  Round 2

Comment 1The article has been revised taking into account the comments.

Response 1:Thank you for your time and valuable comments on our manuscript. We appreciate your guidance and are glad to hear that the revisions align with your suggestions. Please do not hesitate to let us know if any further adjustments are needed.

Reviewer 4 Report

My comments are rather detailed

Line 11, 18 Antibacterial ?????

Line 45  „it”   - capital letter

Line 64-68   The publication of Iantas et al 2021 describes antimicrobial activity of fungi against three fungal pathogens C. abscissum, P. citricarpa and F. graminearum and one bacterial strain X. citri

The authors write : “Considering the need for more effective treatments of plant diseases caused by citrus and maize pathogens, the present study identified fungal endophytes of two medicinal plants, V. divergens and S. adstringens, and investigated their antimicrobial potential against the citrus pathogens C. abscissum, P. citricarpa, and X. citri subsp. citri, as well as against the maize pathogen F. graminearum”  

Pease change the “antibacterial potential” and “antibacterial activity” to antimicrobial

Line 69 again “antibacterial activities”

If the aim of the study was to show activity against Alternaria it is better to underline the antifungal activity of the endophytes.

Line 75    remove one [17]

Line 77  ref. [18] remove superscript

Line 78  …….degrading enzymes in legumes affected by …..

Please read the article [19] carefully. There are no analyses of plants and their enzymatic activity. Experiments are conducted on pathogenic fungi. The authors show that the activity of cellulases and pectinases in pathogenic fungi is inhibited by essential oils from plants.

Line84-89

……often by producing enzymes that break down substances, generating beneficial compounds outside their cells, and assisting plants in strengthening their defenses.[20]… some details are needed look below  

For example, Bacillus spp. are known to control other organisms in a variety of ways, often by producing enzymes that break down fungal cell wall polysaccharides, producing beneficial antimicrobials, enhancing plant growth by synthesizing plant growth hormones, and helping plants strengthen their defenses by inducing systemic plant resistance [20].

Line 91-92 Phytophthora is not a fungus, it belongs to pseudofungi - Oomycota- fungus-like eukaryotic microorganisms

Should be phytopathogenic Oomycete such as …..

Line 144 why authors chose PDA medium to cultivated bacteria. B5 is a bacterial strain.

In Ref 28 Bacillus is cultivated using: yeast extract– 5.0, peptone – 15.0, NaCl – 5.0, distilled water – 1.0 l. To study morphology of the colonies, the same medium was used but with 15 g/l of agar. In addition, ready-to-use MPA (meat-peptone agar) and MPB (meat-peptone broth) were also used.

Line 159 ref [31] remove superscript

Line 170 -172 (previous ver. Line 168) I understand that the concentration of bacteria must be appropriate.   Was the best performing bacteria concentration analyzed by the authors or is it taken from the literature? – ref. is needed

This is an important information since as authors answered “ A concentration that is too low may not be sufficient to inhibit the pathogen, while a concentration that is too high could have adverse effects on plants or the environment.”

Line 231, 254, 295, 561  again “antibacterial activity”…..

Figure 1 Please look at Fig1. Alternaria alternata is black. It seems that in Fig1A the discs are with bacteria that inhibit the growth of Alternaria. Alternaria is only in the center of the plate, but in Fig 1B the discs are with sterile water and Alternaria can grow without any problems.

Section 3.3

fermentation broth of strain B5 (1x108) –to make it clear I will rather add fermentation broth with strain B5

In the previous version I was not sure if the spray contained bacteria or just the medium after the bacteria had been cultured.

Fig 8 please specify what concentration of filtrate was used

Line 468 Artemisia argyi – italic

Line 469 Staphylococcus aureus – italic

Line 474  antifungal activity is against fungi not bacteria !!!!

Line 486 …..strain B5 fermentation solution….  Fig. 5 presented strain B5 fermentation broth with bacteria B5 (1x108) and in the discussion it is not clear what is discussed only fermentation solution or suspension of bacteria.

Line 502 the Ref. 58 is a review about siderophores. See page 200 for information on Bacillus siderophores and their role in the biocontrol of fungal, non-bacterial, diseases.

Line 509  strain B5 – capital letter

Line 541  Ref. 70 remove superscript

Line 561 antifungal activity since authors analyzed antifungal activity not antibacterial.

Author Response

Reviewer #4  Round 2

Comment 1Line 11, 18 Antibacterial ?????

Response 1: Thank you for your careful review. I truly appreciate you pointing out this oversight, and I fully acknowledge this was an oversight on my part. The correction has been made accordingly in the revised manuscript.

Comment 2Line 45  „it”   - capital letter

Response 2: We have carefully revised the manuscript and sincerely appreciate your thorough review. Thank you for identifying this erroryour feedback is greatly valued.

The revised objective now reads: Page 2, lines 45

Comment 3Line 64-68   The publication of Iantas et al 2021 describes antimicrobial activity of fungi against three fungal pathogens C. abscissum, P. citricarpa and F. graminearum and one bacterial strain X. citri

The authors write : Considering the need for more effective treatments of plant diseases caused by citrus and maize pathogens, the present study identified fungal endophytes of two medicinal plants, V. divergens and S. adstringens, and investigated their antimicrobial potential against the citrus pathogens C. abscissum, P. citricarpa, and X. citri subsp. citri, as well as against the maize pathogen F. graminearum 

Please change the antibacterial potential and antibacterial activity to antimicrobial

Response 3: Thank you for your insightful feedback regarding the description of Reference [14]. I have carefully revised the relevant section to correct the key information ensuring it accurately reflects the studys focus on antimicrobial activity against both fungal and bacterial pathogens. Your suggestions have been invaluable in enhancing the accuracy and rigor of the manuscript. Please let me know if any further adjustments are needed.

The revised objective now reads: Page 3 lines 64-70

Comment 4Line 69 again antibacterial activities

If the aim of the study was to show activity against Alternaria it is better to underline the antifungal activity of the endophytes.

Line 75    remove one [17]

Line 77  ref. [18] remove superscript

Response 4: All suggested revisions (terminology adjustment, reference deduplication, and formatting correction) have been carefully implemented. Thank you for your valuable feedback.

Comment 5Line 78  …….degrading enzymes in legumes affected by ..

Please read the article [19] carefully. There are no analyses of plants and their enzymatic activity. Experiments are conducted on pathogenic fungi. The authors show that the activity of cellulases and pectinases in pathogenic fungi is inhibited by essential oils from plants.

Response 5:Thank you for identifying the inaccuracy in my description of Reference [19]. I have revised the sentence to clarify that menthol inhibits cell wall-degrading enzymes secreted by pathogenic fungi (not plant enzymes) in legumes affected by root rot pathogens, as documented in the study. The updated text now correctly reflects the experimental focus on fungal enzyme activity.

The revised objective now reads: Page 3, lines 79-81

Comment 6: Line 84- 89

……often by producing enzymes that break down substances, generating beneficial compounds outside their cells, and assisting plants in strengthening their defenses.[20] some details are needed look below 

For example, Bacillus spp. are known to control other organisms in a variety of ways, often by producing enzymes that break down fungal cell wall polysaccharides, producing beneficial antimicrobials, enhancing plant growth by synthesizing plant growth hormones, and helping plants strengthen their defenses by inducing systemic plant resistance [20].

Response 6:Thank you for suggesting additional details to strengthen the description of Bacillus spp.s mechanisms in Reference [20]. I have revised the sentence to include specific mechanisms documented in the literature, such as producing cell wall-degrading enzymes (e.g., chitinase), synthesizing antimicrobial compounds and plant growth hormones, and inducing systemic plant resistance. These details now accurately reflect the multifunctional roles of Bacillus spp. as described in [20].

The revised objective now reads: Page 3, lines 86-92

Comment 7:  Line 91-92 Phytophthora is not a fungus, it belongs to pseudofungi - Oomycota- fungus-like eukaryotic microorganisms

Should be phytopathogenic Oomycete such as ..

Line 144 why authors chose PDA medium to cultivated bacteria. B5 is a bacterial strain.

Response 7: Thank you for your valuable comments. We have carefully revised the manuscript according to your suggestions.

The revised objective now reads: Page 4, lines 91-92 Page 5, lines 144

Comment 8:Line 159 ref [31] remove superscript

Line 170 -172 (previous ver. Line 168) I understand that the concentration of bacteria must be appropriate.   Was the best performing bacteria concentration analyzed by the authors or is it taken from the literature? ref. is needed

Response 8:Thank you for your valuable suggestion regarding the bacterial concentration. The concentration of 1×10⁸ CFU/mL used in the study was supported by [(Meng et al.,2024], which demonstrated no significant difference in control efficacy between this concentration of B4-7 fermentation broth and streptomycin in the results section. Additionally, this concentration aligns with the standard range commonly used in microbiology research for ensuring experimental reproducibility. The reference has been appropriately inserted in the text,please let us know if further adjustments are needed.

Comment 9: Figure 1 Please look at Fig1. Alternaria alternata is black. It seems that in Fig1A the discs are with bacteria that inhibit the growth of Alternaria. Alternaria is only in the center of the plate, but in Fig 1B the discs are with sterile water and Alternaria can grow without any problems.

Response 9: Thank you for your valuable comment regarding Figure 1. We appreciate your careful observation and have revised the figure legend accordingly to better reflect the experimental results. Your suggestion has helped improve the clarity of our presentation.

Additionally, I have re-inoculated the plates, and the mycelia are still in the growth stage, so it may be necessary to wait a few more days. I did not obtain the experimental results within five days, but I can send them to you as an attachment at a later stage. Thank you for your valuable comments, which we will earnestly adopt.

Comment 10: Section 3.3

fermentation broth of strain B5 (1x108) to make it clear I will rather add fermentation broth with strain B5.In the previous version I was not sure if the spray contained bacteria or just the medium after the bacteria had been cultured.

Response 10: Thank you for your meticulous and constructive feedback. We appreciate your attention to clarity and have revised the text in Section 3.3 to:

"fermentation broth with strain B5 (1×10⁸ CFU/mL)"to explicitly indicate the presence of viable bacteria. This modification ensures unambiguous interpretation of the methodology.

The revised objective now reads: Page 5, lines 169-175 Page 12, lines 347-349

Comment 11: Fig. 8, please specify what concentration of filtrate was used Line 468 Artemisia argyi italic Line 469 Staphylococcus aureus italic Line 474  antifungal activity is against fungi, not bacteria !!!

Response 11: Thank you for your valuable comments. We have carefully implemented all the suggested revisions throughout the manuscript, including the specification of filtrate concentration in Figure 8, proper italicization of species names, and correction of the antifungal activity description.

Comment 12: Line 486 ..strain B5 fermentation solution.  Fig. 5 presents strain B5 fermentation broth with bacteria B5 (1x108), and in the discussion, it is not clear whether it is only the fermentation solution or the suspension of bacteria.

Response 12: Thank you for your valuable feedback. We sincerely appreciate your careful reading and constructive suggestions regarding the terminology consistency between the text and Figure 5.

To address your concern, we have revised the description in Line 486 and throughout the manuscript to consistently use "fermentation broth with strain B5 (1×10⁸ CFU/mL)" to explicitly indicate that the material contains viable bacterial cells. This modification ensures clarity and aligns with the presentation in Figure 5. The revised objective now reads: Page 21, lines 485-488

Comment 13:Line 502, the Ref. 58 is a review about siderophores. See page 200 for information on Bacillus siderophores and their role in the biocontrol of fungal, non-bacterial diseases.

Response 13: Thank you for your careful reading and constructive comment. We sincerely appreciate you pointing out this oversight.Your attention to detail has significantly improved our manuscript's precision.

Comment 14:Line 509  strain B5 capital letter; Line 541  Ref. 70 remove superscript; Line 561 antifungal activity since the authors analyzed antifungal activity, not antibacterial.

Response 14:Thank you for your meticulous review and the constructive feedback, which have been invaluable in ensuring the rigor of my manuscript. I have taken careful note of the issues you raised and will be sure to address them thoroughly in future revisions. Please do not hesitate to let me know if you identify any further concerns your insights are greatly appreciated and will help strengthen the quality of the work.

The revised objective now reads: Page 23, lines 561

Round 3

Reviewer 4 Report

I have no major comments

Line 91 and 98 Bacillus - italic